# Thalamus enables active dendritic coupling of inputs arriving at different cortical layers

Arco Bast [1,2,4,7], Jason M. Guest[1,2,5,7], Rieke Fruengel [1,2], Rajeevan T. Narayanan[1,6], Christiaan P. J. de Kock [3] & Marcel Oberlaender [1,3] ✉

Dendritic calcium action potentials (APs) enable the main output neurons of the cerebral cortex – pyramidal tract neurons (PTs) – to associate inputs that arrive at different cortical layers. How synaptic inputs evoke calcium APs during in vivo conditions is yet unclear. We combine in vivo recordings in male rats with synaptic input reconstructions, multi-scale modelling and optogenetic manipulations. We find that thalamocortical (TC) synapses, which provide sensory input to cortex, target specifically and most densely the dendritic domain that initiates calcium APs in PTs. Sensory input from thalamus is hence a reliable, but weak source for activating the dendritic calcium domain. Because it is fast and local, this activation enables active dendritic coupling of sensory input with multiple sensory-evoked and ongoing input streams that arrive during and surprisingly before the stimulus. This 'TC coupling' mechanism accounts for the modulation of the first sensory responses that leave the cortex with bursts of APs.

The cerebral cortex continuously transmits information to subcortical regions. These descending cortical output streams drive interactions between cortex and subcortical regions that are crucial for cognitive processes and for generating behavior[1–3]. Theoretical studies showed that short bursts of high-frequency action potentials (APs) could form a multiplexed neural code that would enable neurons to transmit simultaneously information from multiple input streams[4]. Experimental studies showed that bursts in descending cortical output enhance interactions between cortex and subcortical regions[5–9]. Recent studies provided evidence that the transmission of multiplexed information to subcortical regions via bursts, as well as the resulting interactions with subcortical regions, link descending cortical output streams directly to the perceptual process[10–12].

Pyramidal tract neurons (PTs) in cortical layer 5 (L5) are the major source of descending axons to subcortical regions[13]. Along elaborate dendrites, PTs combine multiple input streams that arrive at all cortical layers[13]. Short bursts are a common firing mode of PTs[14–17]. We recently showed that increased firing of short bursts distinguishes the sensory-evoked responses of PTs from those of all other excitatory neurons in the cortical column[15]. In support of a multiplexed neural code, bursts encoded more information about the stimulus than the PTs' firing rates[15]. Taken together, bursts in sensory-evoked responses of PTs may transmit multiplexed information from all cortical layers, and thereby drive interactions with subcortical regions that enhance sensory processing[7].

A mechanism that enables PTs to combine inputs that arrive at different cortical layers into bursts was discovered in vitro and is referred to as coincidence detection[16]. Back-propagating APs (bAPs) depolarize the initiation zone for calcium-dependent plateau potentials – Ca²⁺ APs – in the apical dendrites of PTs, which opens a short time window for distal dendritic inputs to drive bursts. However, in vivo studies suggest that bAPs have little influence on apical

[1]In Silico Brain Sciences Group, Max Planck Institute for Neurobiology of Behavior – caesar; Ludwig-Erhard-Allee 2, Bonn, Germany. [2]International Max Planck Research School for Brain and Behavior; Ludwig-Erhard-Allee 2, Bonn, Germany. [3]Department of Integrative Neurophysiology, Center for Neurogenomics and Cognitive Research, VU Amsterdam; De Boelelaan 1100, Amsterdam, the Netherlands. [4]Present address: Janelia Research Campus, Howard Hughes Medical Institute, Ashburn, USA. [5]Present address: Functional Architecture and Development of Cerebral Cortex, Max Planck Florida Institute for Neuroscience; 1 Max Planck Way, Jupiter, Florida, USA. [6]Present address: Department of Neurology, University of Düsseldorf, Düsseldorf, Germany. [7]These authors contributed equally: Arco Bast, Jason M. Guest. ✉e-mail: marcel.oberlaender@mpinb.mpg.de

dendritic Ca²⁺ signals[11,12,18,19]. It is hence unclear whether bursts in sensory-evoked PT output originate via coincidence detection. Indeed, it was speculated that the sensory-evoked Ca²⁺ signals in the apical dendrites of PTs, and hence bursts in sensory-evoked cortical output, could originate from a local activation of the Ca²⁺ zone[12]. Circuits that could provide such a local activation remain unknown. Moreover, PTs may generate sensory-evoked bursts even without Ca²⁺ electrogenesis in the apical dendrite[20]. Therefore, here we ask: *Which mechanisms underlie bursts in sensory-evoked cortical output?*

We hypothesize that the sensory input, which PTs receive directly from the thalamus, could be an origin of bursts in sensory-evoked cortical output. The primary thalamus is the main route by which sensory information reaches cortex[21–23]. These thalamocortical (TC) axons terminate densely in L4 and at the L5-L6 border[24,25], but form synapses with virtually all types of neurons and across all cortical layers[26–28]. It was even suggested that direct TC input could drive the sensory-evoked responses of PTs[29]. We therefore systematically reconstructed the distributions of synapses from the primary thalamus along the dendrites of in vivo recorded PTs. Consistent with studies in vitro[28,30], we show that TC axons target both the basal and apical dendrites of PTs. However, surprisingly, we find that TC synapses cluster most densely near the dendritic initiation zone for Ca²⁺ APs. The synchronous volley of sensory-evoked input that PTs receive directly from the thalamus[29,31] could hence contribute to both the generation of somatic APs and dendritic Ca²⁺ APs. It seems, therefore, likely that TC input to PTs facilitates the modulation of sensory-evoked cortical output with bursts.

One challenge for testing this hypothesis is the fact that neurons of virtually all types in the cortical column can respond near simultaneously to sensory stimuli[14,15]. Essentially all of these types are presynaptic to PTs[26,28,32,33]. Consequently, PTs receive direct sensory input from the thalamus[29] and multiple streams of indirect sensory input from TC-driven cortical neurons[32]. Approaches that could dissect how multiple input streams interact across all dendritic domains remain limited to in vitro studies. Even if this technical limit was resolved, the enormous computational complexity of their dendrites poses another challenge[34] for inferring causality between the inputs that PTs receive in vivo and their AP outputs[35]. Thus, the mechanisms underlying bursts in sensory-evoked cortical output may only be revealed if the dendritic locations, strengths and activation times of all inputs from the thalamus and cortex are known, and integration of these input patterns could be studied with respect to the complex properties of the PTs' dendrites.

We tackle these challenges by combining in vivo recordings and neuroanatomical reconstructions with multi-scale simulations[32]. We record sensory-evoked APs of PTs, reconstruct TC synapses along their dendrites, and convert these in vivo recorded PTs into neuron models that capture their biophysical (e.g., Ca²⁺ APs) and cellular mechanisms (e.g., coincidence detection). We embed the PT models into network models of the vibrissal area in the rat primary somatosensory cortex (vS1) – the barrel cortex[36]. The network models provide validated constraints for which neurons in the thalamus and barrel cortex provide input to PTs and where on the dendrites these inputs occur[32]. We activate the network models with recording data from the thalamus and barrel cortex, acquired for the same in vivo condition for which we seek to explain PT activity. These multi-scale models thereby enabled us to study in silico how the dendrites of in vivo recorded PTs could in principle, transform in vivo-like input patterns into their observed AP outputs.

The simulations predicted burst firing as observed in vivo, and identified a strategy for testing the underlying mechanism, which we did in vivo via manipulations of the same PTs that we simulated. We find that direct sensory input from the thalamus is a source for the local activation of the Ca²⁺ zone in the apical dendrites of PTs, which enables these cortical output neurons to transmit bursts that multiplex sensory with ongoing information streams. We term this mechanism 'TC coupling'.

## Results

### Anatomical measurements to constrain simulations of sensory-evoked cortical output

We quantified the distributions of TC synapses along the dendrites of in vivo recorded PTs whose sensory-evoked responses we seek to explain in silico (Fig. 1a). For this purpose, we injected an adeno-associated virus (AAV) into the thalamus (N = 26 rats). The virus (rAAV2/1-Syn-hChR2-mCherry) spreads in the anterograde direction along axons of infected neurons, where it expresses the light-gated ion channel Channelrhodopsin (ChR2) and the red fluorescent protein mCherry in presynaptic terminals. We targeted our injections to the ventral posterior medial nucleus (VPM), the vibrissal region of the primary thalamus. Reconstructing the volume of fluorescent labeling in the thalamus showed that the injection sites were located within the VPM (Supplementary Fig. S1a), that the center locations of the injections varied by less than ± 58 μm across rats (Supplementary Fig. S1b), and that the virus did not spread into the posterior medial nucleus (POm) of the higher-order thalamus (Supplementary Fig. S1c). The spread of the virus within the barrel cortex supports this conclusion (Supplementary Fig. S1d). We did not observe density peaks of fluorescent labeling in L1 or at the L4-L5 border, the major target sites of POm axons[25]. Instead, labeling was densest in L4 and at the L5-L6 border, the major target sites of VPM axons[25,26]. Both the vertical distribution of the virus across layers and the horizontal distribution of the virus across the barrel field matched those obtained by single VPM axons (Supplementary Fig. S1e). Thus, our injections enabled us to anatomically quantify and to optogenetically activate TC synapses in the barrel cortex that originate from the primary thalamus of the vibrissal system.

In virus-injected anesthetized rats (N = 26), we used cell-attached recording pipettes to in vivo label individual neurons (n = 62) in the barrel cortex with Biocytin for *post hoc* reconstruction (Supplementary Fig. S2a) and cell type classification[26,37]. In 18 rats, we identified 25 of the neurons as PTs (Supplementary Fig. S2b). The remaining 37 neurons belonged to other excitatory types – spiny neurons in L4 (L4SP, n = 4), intratelencephalic neurons in L5 (L5IT, n = 5) and corticocortical neurons around the L5-L6 border (L6CC, n = 6) – or inhibitory types (n = 22). We selected 10 PTs with high-quality labeling (Supplementary Fig. S2a) for quantification of TC synapses along their dendrites (Supplementary Fig. S2c). For this purpose, we immunolabeled their brain slices with antibodies against the vesicular glutamate transporter 2 (VGlut2), which is expressed specifically in TC synapses[38]. We identified all contact sites between Biocytin-labeled dendritic spines and virus-infected axonal boutons, and then inspected via super-resolution light microscopy whether contact sites co-localized with VGlut2 (Fig. 1b). We hence quantified the distributions of TC synapses from the VPM along the dendrites of in vivo recorded PTs (n = 10, 13,296 contacts, median fraction of VGlut2-positive contacts: 86%). For comparison, we also quantified TC synapses along the dendrites for one L4SP, L5IT and L6CC (Supplementary Fig. S2d).

Around 60% of the TC synapses on PTs were located along the basal and apical oblique dendrites, and hence proximal to the soma (Supplementary Fig. S2e). This innervation of the proximal dendrites was well-predicted by overlap[26] between single VPM axons and the PTs' dendrites (Supplementary Fig. S2f). For the L4SP, L5IT and L6CC, axodendritic overlap even predicted the TC synapse distributions along all their dendritic domains (Supplementary Fig. S2g). However, TC synapse distributions were not predictable by axo-dendritic overlap for the apical trunks of PTs (Fig. 1c), where TC synapses accumulated more densely than expected (Supplementary Fig. S2e, f). In fact, TC synapse densities increased along the trunk with soma distance, reached a peak ~100 μm below the primary branch point (BP), and then

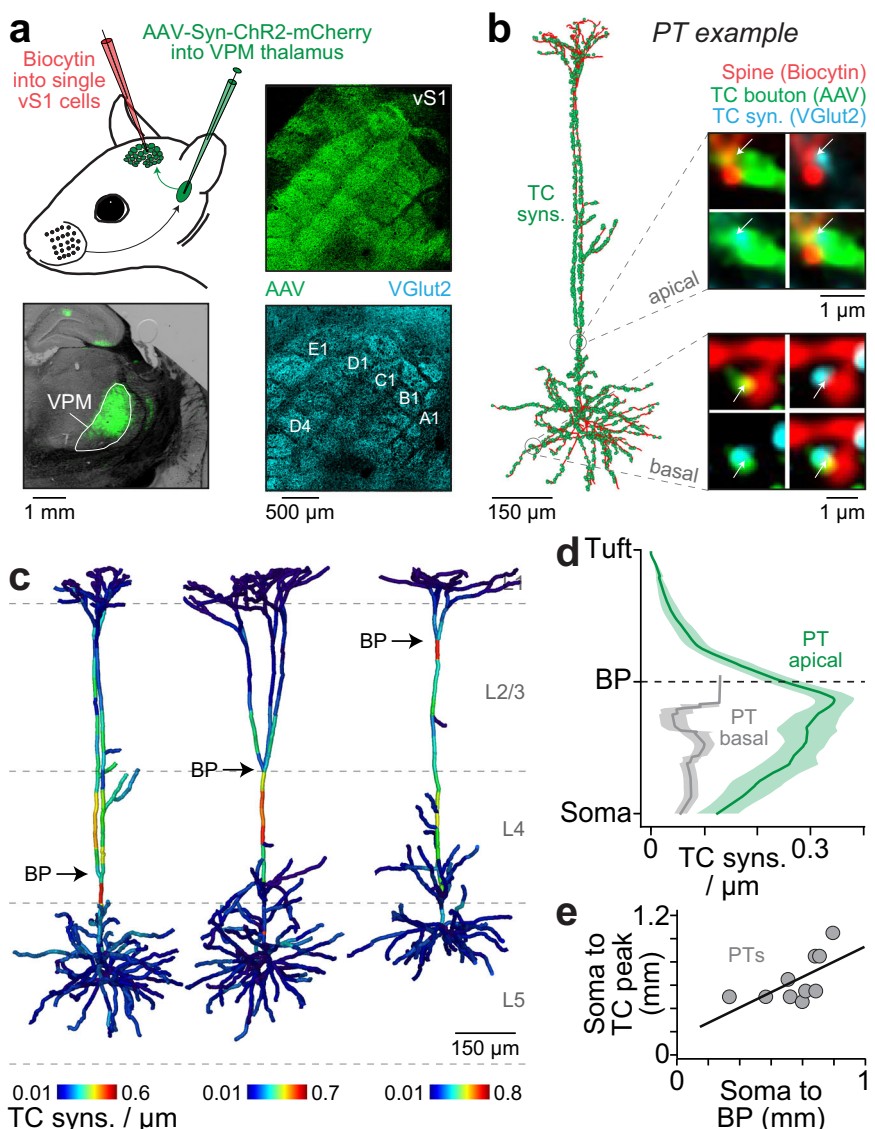

**Fig. 1 | Distribution of TC synapses along the dendrites of in vivo recorded PTs.**
**a** Top-Left: we labeled in vivo recorded neurons in the barrel cortex (vS1) with Biocytin in adeno-associated virus (AAV) injected anesthetized rats ($N = 26$). Bottom-left: AAV injection example into ventral posterior medial thalamic nucleus (VPM). The schematic of the rat was adapted and modified with permission from "Rhythmic whisking by rat: retraction as well as protraction of the vibrissae is under active muscular control. Berg RW & Kleinfeld D. J Neurophysiol 89, 2003, page 105."[67]. Top-right: image of the barrel field in layer 4 of vS1 shows AAV-infected thalamocortical (TC) axons corresponding to the injection site on the left. Bottom-right: expression of the vesicular glutamate transporter 2 (VGlut2) corresponding to the top panel image. **b** 3D reconstruction of dendrites (red) and TC synapses (green) of the PT example shown in Supplementary Fig. S2a. Zoom-ins show super- resolution images of dendritic spines (red) on the basal (bottom) and apical dendrites (top) that were in contact with AAV-infected TC boutons (green) and which co-expressed VGlut2 (cyan). The arrows denote the locations of the hence identified TC synapses. **c** Density of TC synapses along the dendrites of three PT examples (see Supplementary Fig. S2c for complete gallery). **d** We aligned PTs by the BP and plotted TC synapse densities (mean ± SEM) along the basal and apical oblique dendrites (gray), and the apical trunk and tuft dendrites (green). **e** BP locations correlated significantly with the location of the highest TC synapse density (Pearson $R = 0.8$, $p = 0.005$). ***Please note:*** we chose the color maps in panels a/b to increase visibility. Fluorescent labeling was Alexa-405 for VGlut2, Alexa-488 for biocytin and Alexa-647 for AAV. Source data for panel (**d**) are provided in the Source Data file.

decayed exponentially towards the tufts (Fig. 1d). Strikingly, this innervation pattern resembles the pattern of $Ca^{2+}$ influx into PTs, which also increases towards a peak near the primary BP[39]. Axons from the primary thalamus may hence target specifically the dendritic domain of PTs that initiates $Ca^{2+}$ APs[39,40]. Supporting this interpretation of the density peak, we show that the specific innervation of the primary BP by TC synapses is observed across PTs (Fig. 1e), irrespective of BP location in L2, L3 or L4 (Supplementary Fig. S2a). Moreover, the density peak did not reflect higher spine densities near the primary BP, but instead a 3-fold higher fraction of spines with TC synapses (Supplementary Fig. S2h). Overall, our anatomical findings indicate that direct

sensory input from the thalamus could be a source for the local activation of $Ca^{2+}$ channels in the PTs' apical trunks.

## Electrophysiological measurements to constrain simulations of sensory-evoked cortical output

Prior to the labeling of the PTs with Biocytin, we had recorded their AP patterns in response to a standard stimulus[41,42], an airpuff that deflects the vibrissae (i.e., whiskers) in the caudal direction (Fig. 2a). PTs ($n = 25$) responded to the multi-whisker deflection with three types of patterns: 1 AP or short bursts of 2 or 3 APs (Fig. 2b). The onset times of 1 AP responses did not differ from those of bursts

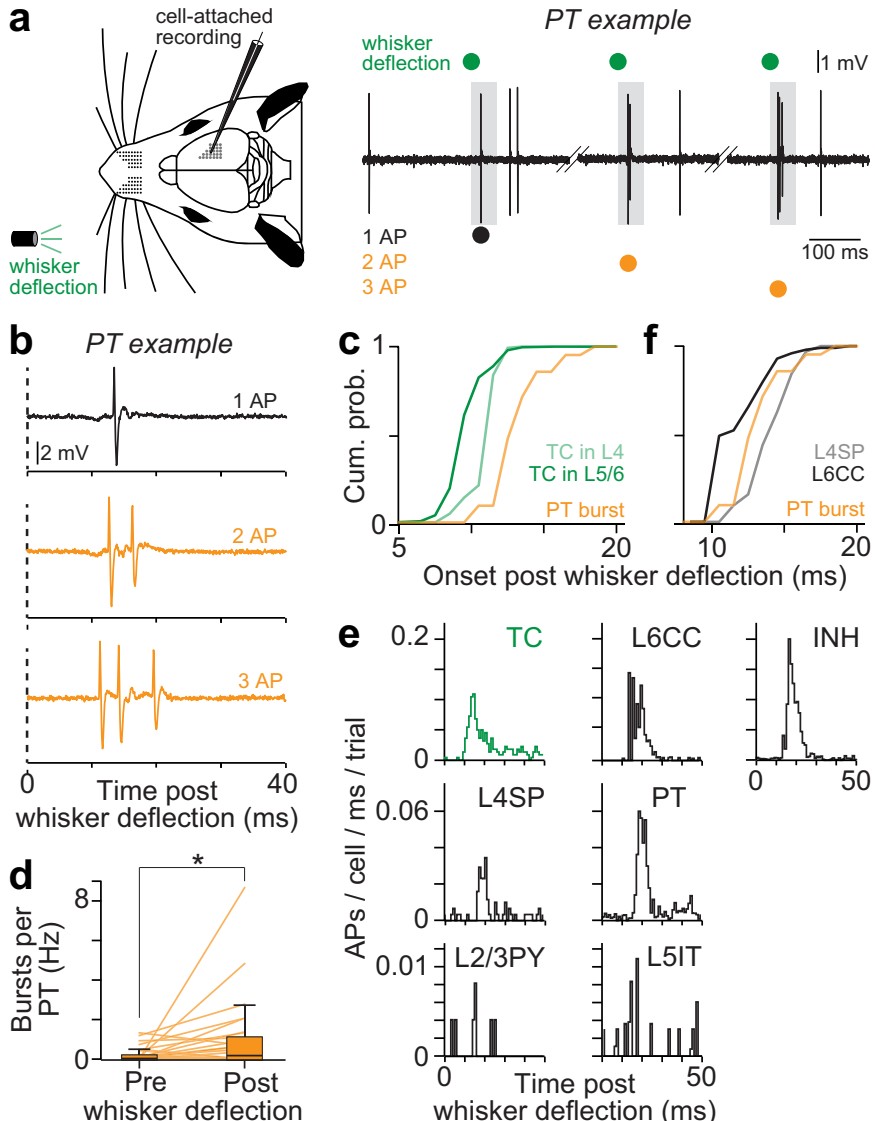

**Fig. 2 | Bursts in sensory-evoked cortical output by PTs. a** Left: schematic of our experiments in anesthetized rats. The schematic of the rat was adapted and modified from "Diamond ME, von Heimendahl M, Knutsen PM, Kleinfeld D, Ahissar E. 'Where' and 'what' in the whisker sensorimotor system. Nat Rev Neurosci 9, page 602, 2008, Springer Nature"[68]. Reproduced with permission from Springer Nature. Right: example recording for one of the PTs in Fig. 1c (most right). The trace shows somatic APs recorded during four example trials in which we deflected the whiskers by a low-pressure airpuff. Gray boxes denote the first 50 ms after stimulus onset for which we analyzed sensory-evoked responses. **b** Example trials for the PT from panel a show three types of sensory-evoked responses. **c** Latencies of sensory-evoked local field potentials (LFPs) show that TC input reaches L5/6 and L4 (n = 62 recording depths) before the onsets of sensory-evoked bursts by PTs. **d** Burst rates pre and post whisker deflections (n = 25 PTs). Box plots represent medians and 25th to 75th percentiles. Whiskers extend to 1.5 times the interquartile range. The asterisk represents 2-sided t Test, unpaired, heteroscedastic, p = 0.026). **e** Post-stimulus-time-histograms (PSTHs) for TC neurons in VPM (n = 8), L2/3PNs (n = 8), L4SPs (n = 10), L5ITs (n = 9), PTs (n = 25), L6CCs (n = 7) and L2-6 INHs (n = 22). **f** Latencies of sensory-evoked APs in L6CCs, L4SPs and PTs. Source data for panels (c–f) are provided in the Source Data file.

(2-sided t test, paired, p = 0.78). Across trials, PTs hence elicited their first sensory-evoked APs always at the same time, irrespective of whether the response occurred as 1 AP or bursts of 2 or 3 APs (Fig. 2c). In contrast, the relative occurrence of the three response types varied between PTs. We observed 1 AP responses most frequently (89% of the trials) and in all 25 PTs, followed by bursts of 2 APs (8%), which occurred in 13 PTs. Bursts of 3 APs occurred in 8 PTs and were least abundant (3%). However, in PTs with bursts, patterns of 2 or 3 APs accounted for up to 40% (median: 10%) or 14% (median: 5%) of the responses, respectively. Indeed, compared to periods of ongoing activity (Fig. 2d), bursts increased significantly in PTs upon stimulus onset (Wilcoxon rank-sum test: p = 0.016). Thus, we were able to investigate the origins of burst firing in sensory-evoked

cortical output for our experimental condition of multi-whisker deflections in anesthetized rats.

We quantified how populations that are presynaptic to PTs could contribute to their responses. For this purpose, we recorded AP patterns of 89 identified neurons in the VPM thalamus and barrel cortex. VPM neurons, as well as excitatory and inhibitory neurons across all layers of the barrel cortex, responded to multi-whisker deflections (Fig. 2e). Among excitatory neurons, L6CCs responded most reliably, followed by PTs and L4SPs. Responses of pyramidal neurons (PNs) in L2/3 and of L5ITs were nearly absent. Responses of L6CCs preceded those of PTs (Fig. 2f), whereas responses of L4SPs occurred near simultaneously with those of PTs (1-way ANOVA: p = 0.001 for L6CC; p = 0.9 for L4SP). The delay between L6CC and L4SP responses, on

average 2.1 ms ($p = 0.0023$), is in line with delays due to conductance velocity and branching patterns of VPM axons[32,43]. Local field potential (LFP) recordings support this interpretation as they reveal delays between sensory-evoked TC input to L5/6 and L4 (Fig. 2c), on average 1.8 ms ($n = 735$ trials, N = 17 rats, 2-sided $t$ test, $p = 2.5 \times 10^{-32}$). Thus, prior to their first sensory-evoked APs, PTs could receive sensory input from VPM thalamus in three ways: indirectly via L6CCs, directly via the low density but high number of TC synapses along the basal dendrites, and directly via the high density of TC synapses along the apical trunk. After their first sensory-evoked APs, and throughout the duration of bursts, PTs could additionally receive sensory-evoked input via recurrent connections from PTs, from L4SPs, and to lesser degrees from L5ITs and L2/3 PNs. In essence, these results indicate that bursts in sensory-evoked cortical output originate (also during our experimental condition) from direct and multiple indirect sensory input streams that impinge onto all dendritic domains of PTs.

## Multi-scale simulations of sensory-evoked cortical output

How do bursts in sensory-evoked cortical output originate in the dendrites of PTs from interactions between multiple sensory input streams? How does direct sensory input from the thalamus contribute to these interactions, in particular, the dense TC input to the dendritic $Ca^{2+}$ domain near the primary BP? We used multi-scale simulations to address these questions (Supplementary Movie 1). For this purpose, we converted the dendrites with reconstructed TC synapses of in vivo recorded PTs into biophysically-detailed models (Supplementary Fig. S3a–c). We selected the PTs with the most superficial and deepest BPs, and one from the bulk of the distribution. For each morphology, we identified a large number of models ($n = 909,722 / 292,151 / 201,028$), which captured somatic, dendritic and axonal electrophysiology of PTs as observed for different current injections in vitro[44], but used very different biophysical parameters to do so (Supplementary Fig. S3d). We accounted for this degeneracy of biophysical parameters by selecting models for each morphology ($n = 40 / 15 / 13$) that covered the parameter ranges for each ion channel (Supplementary Fig. S3e), as well as different biophysical mechanisms to generate $Ca^{2+}$ APs (Supplementary Fig. S3f). We then simulated unitary postsynaptic potentials (uPSPs) evoked at the soma by each of the reconstructed TC synapses (Supplementary Fig. S3g), and varied their strengths until uPSP distributions in silico matched with those observed in vivo[29]. The hence generated set of PT models ($n = 68$) captured morphological, electrophysiological and biophysical diversity of PTs, as well as variability in the number, dendritic distribution and strength of TC synapses from the VPM.

We embedded each PT model into anatomically detailed and empirically validated network models of the rat barrel cortex[36]. These embeddings generated connectivity patterns that captured which neurons in the VPM thalamus and across all layers of the barrel cortex could provide input to these in vivo recorded PTs (Fig. 3a), and where along the dendrites these inputs could occur (Fig. 3b). As for TC synapses, we simulated uPSPs that synapses of neurons in the barrel cortex evoke at the soma, and varied their strengths until the uPSP distributions in silico matched for each type with those observed in vitro[45]. We repeated the embeddings to generate different, yet equally realistic, anatomical and functional connectivity patterns ($n = 9720$) for each PT model ($n = 68$). The hence extended set of 660,960 PT models (i.e., $9720 \times 68$) captured diversity of PTs, variability in TC input, but now also variability in the number, dendritic distribution and strength of synapses from excitatory and inhibitory types across all layers of the barrel cortex (Supplementary Fig. S4a–c).

We used our recording data from the VPM thalamus and barrel cortex (Fig. 2e) to activate neurons in the network models, and hence synapses along the dendrites of the PT models. We generated such input patterns that mimic our experimental condition for periods of ongoing activity ($n = 440,648$) and during multi-whisker deflections

($n = 5,732,274$). We then simulated how the dendrites of each PT model transform these synaptic inputs into somatic output (Fig. 3c). Without tuning any of the parameters, about 10% of the PT models ($n = 67,424$) predicted ongoing and sensory-evoked firing rates as we recorded in vivo, including the three response types (Fig. 3d). Even though we did not select models to match this in vivo observation, 1 AP responses occurred also most frequently in silico, followed by bursts of 2 APs, whereas bursts of 3 APs were least abundant (Fig. 3e). PT models that predicted our in vivo recordings comprised all dendrite morphologies, all biophysical parameter sets and all connectivity patterns. Thus, our simulations enabled us to explore possible origins of bursts in sensory-evoked cortical output, and to test whether models that capture PT diversity from subcellular to network scales agree about the origins. For this purpose, we introduce the term 'model consensus' for in silico results that generalize across our diverse set of PT models.

## In silico manipulations of sensory-evoked cortical output

We replayed the simulations that predicted our in vivo recordings and performed manipulations at (sub)cellular and circuit levels. First, we investigated how direct sensory input from the thalamus contributes to the response types of PTs by performing two in silico manipulations. We deprived all PT models from all indirect sensory input via TC-driven excitatory neurons in the barrel cortex (manipulation 1), or from direct sensory input via TC synapses along their dendrites (manipulation 2). Manipulation 1 abolished AP responses in virtually all multi-scale models (Fig. 4a). In contrast, manipulation 2 abolished specifically all bursts (Fig. 4b) – i.e., PTs maintained sensory-evoked responses, but responded with 1 AP instead of bursts (Fig. 4a). These manipulations provided model consensus that both direct and indirect sensory inputs are required for generating bursts in sensory-evoked cortical output (Supplementary Fig. S5a–c). However, we predict that direct sensory input alone is too weak to drive responses, whereas indirect sensory input alone can drive responses, but only with 1 AP. In essence, these results indicate that without direct sensory input from the thalamus, PTs would still respond to stimuli, but not with bursts.

We performed additional in silico manipulations in which we deprived PTs from indirect sensory input by each cortical excitatory type separately (manipulation 3–7). These manipulations showed that indirect sensory input via L6CCs (manipulation 7) is most critical for evoking sensory-evoked APs in PTs (Supplementary Fig. S5d). In fact, when we deprived PTs from L6CC input, as well as from the direct sensory input (manipulation 8), AP responses were abolished in virtually all multi-scale models (Supplementary Fig. S5e). In line with our previous reports[32,35], these results show that sensory-evoked cortical output is driven primarily by interactions between direct sensory input from the TC→PT and indirect sensory input from the TC→L6CC→PT pathways. In further support of this conclusion, all PT models generally maintained their sensory-evoked responses when we deprived them from indirect sensory input (manipulation 3–6) by any of the other excitatory types (Supplementary Fig. S5d). However, manipulations of each type affected to varying degrees burst occurrences, i.e., PTs responded with 1 AP instead of bursts (Supplementary Fig. S5f, g). Overall, these manipulations provided model consensus that interactions between direct and indirect sensory input via L6CCs drive sensory-evoked cortical output with 1 AP, but that a modulation of these responses with bursts results from additional interactions of direct sensory input with virtually all indirect sensory input streams.

Where direct and indirect sensory inputs interact on the dendrites differs between excitatory types (Supplementary Fig. S4a–c). For example, L6CCs target primarily basal dendrites, L4SPs apical dendrites of PTs[37]. We performed two in silico manipulations to investigate these differences. We deprived all PT models from the active biophysical properties of their apical dendrites (manipulation 9), or from direct sensory input at different distances from the soma

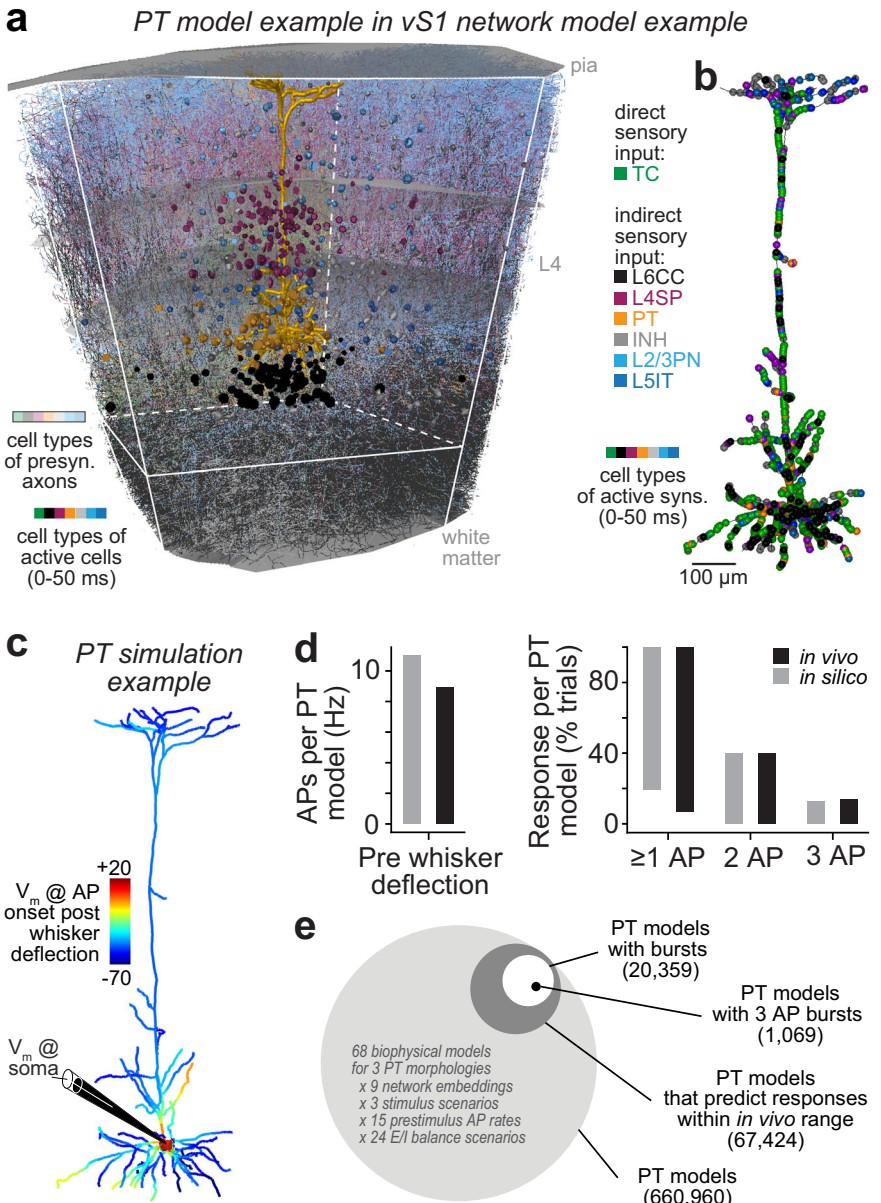

**Fig. 3 | Simulations predict sensory-evoked cortical output as observed in vivo.** **a** Example of a multi-scale model. PT whose anatomy and physiology are shown in Fig. 1c and Fig. 2a, b is converted into biophysically-detailed multi-compartmental models (Supplementary Fig. S3) and embedded into our network model of vS1[36]. Shaded colors represent axons for one configuration of neurons that are pre-synaptic to the PT. Solid colors represent somata for one configuration of these presynaptic neurons that provide input after stimulus onset. **b** Example of distribution of active synapses along the dendrites corresponding to the configuration of active cells in panel (**a**). **c** Example simulation shows membrane potential (V$_m$) as predicted for passive whisker deflection corresponding to the configuration of active synapses in panel (**b**). **d** We simulated how PTs transform synaptic inputs that mimic our experimental condition into somatic APs. About 10% of the models (see panel **e**) predicted pre- and post-stimulus activity, as we had observed in vivo. **e** We generated 660,960 multi-scale model configurations of PTs, which captured morphological, electrophysiological and biophysical diversity of PTs, as well as variability of input from VPM and vS1. Source data for panel (**d**) are provided in the Source Data file.

(manipulation 10). Strikingly, passive apical dendrites abolished bursts of 3 APs, but not bursts of 2 APs (Fig. 4a, b). Similarly, reducing direct sensory input to apical dendrites, particularly around the primary BP, abolished bursts of 3 APs, but not bursts of 2 APs (Fig. 4c). In contrast, reducing direct sensory input to basal dendrites abolished bursts of both 2 and 3 APs. These manipulations provided model consensus that bursts of 2 and 3 APs originate from different interactions of direct with indirect sensory input streams. Both types of bursts require interactions in basal dendrites, but bursts of 3 APs require additional interactions in apical dendrites, as well as the active properties of this dendritic domain.

What happens in the apical dendrites that distinguishes bursts of 3 APs from the other response types? We addressed this question by quantifying the simulated membrane potentials at the Ca$^{2+}$ domain near the primary BP (Fig. 5a). In all PT models, sensory input triggered reliably dendritic potentials, which differed, however, in their waveforms (Fig. 5b). In the vast majority of the simulations (97% of 8,551,091 trials), sensory input failed to elicit dendritic Ca$^{2+}$ APs. Instead, a fast rise and a quick decay phase characterized the waveform of these most frequently occurring sensory-evoked dendritic potentials (Fig. 5b, **black trace**). In the remaining trials, sensory input evoked dendritic Ca$^{2+}$ APs that rose equally fast but decayed

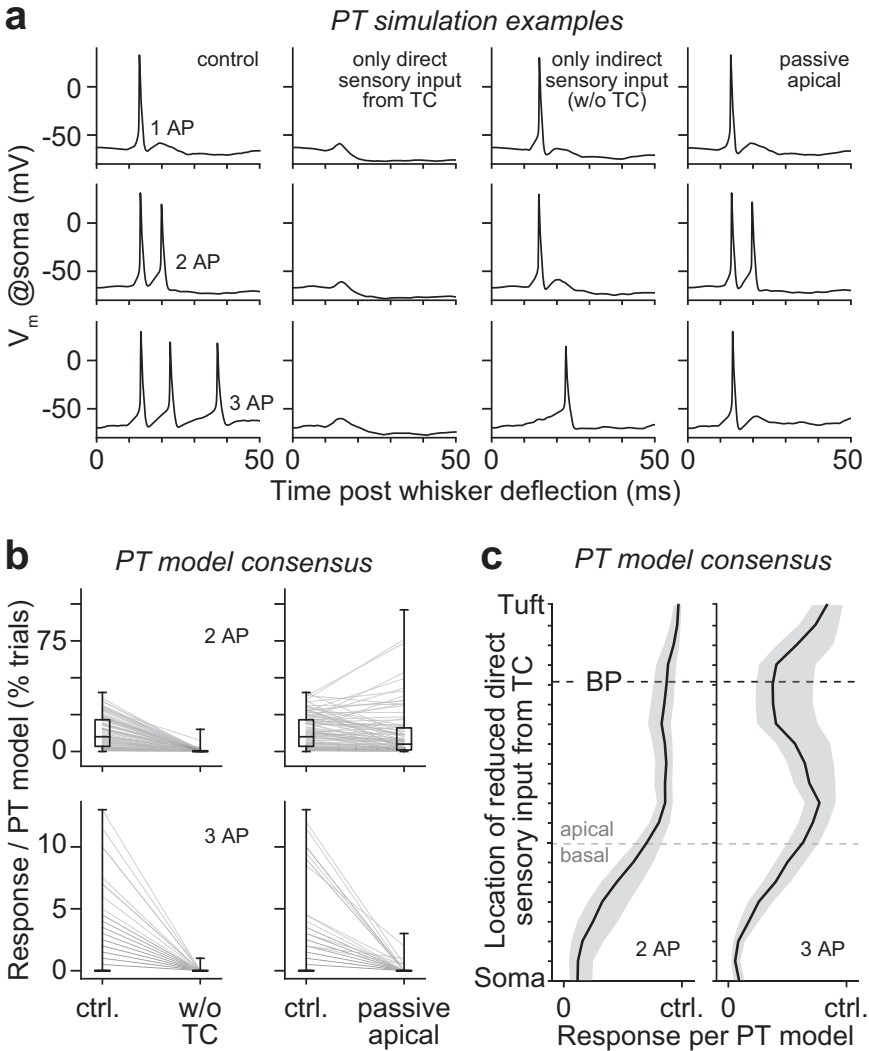

**Fig. 4 | In silico manipulations of sensory-evoked cortical output. a** Simulation examples for the example model in Fig. 3a–c show how in silico manipulations affect sensory-evoked responses with 1 AP (top row) and bursts of 2 (center) or 3 APs (bottom). From left to right: no manipulation (i.e., control), we deprived PTs from any indirect sensory input via TC-driven neurons in vS1 (i.e., responses are evoked solely by direct sensory input via the TC→PT pathway; manipulation 1), we deprived PTs from any direct sensory input via the TC→PT pathway (i.e., responses are evoked solely indirect sensory input via TC-driven neurons in vS1; manipulation 2), we deprived PTs from the active properties of their apical dendrites (i.e., PTs could not generate $Ca^{2+}$ APs; manipulation 9). **b** Model consensus from all 20,359 model configurations which exhibit burst responses: both types of bursts require

direct sensory input from the thalamus (left), whereas bursts of 3 APs, but not bursts of 2 APs, require the active properties of the apical dendrites (right). Box plots represent medians and 25th to 75th percentiles. Whiskers extend to the full range of the data. **c** Model consensus from manipulation 10 (i.e., we deprived PT models from direct sensory input at different distances from the soma): both types of bursts require direct sensory input to basal dendrites, whereas bursts of 3 APs, but not bursts of 2 APs, also require direct sensory input to apical dendrites – in particular around the primary BP. Shadings represent the 25th to 75th percentiles, bold lines represent the medians. Source data for panels (**b**, **c**) are provided in the Source Data file.

much slower with several peaks and dips (Fig. 5b, **orange trace**). Our in silico predictions are supported by in vivo dendritic recordings for the same experimental condition – i.e., for passive whisker deflections in anesthetized rats[46]. In fact, the in silico predicted durations and amplitudes of both the fast potentials and the $Ca^{2+}$ APs were virtually indistinguishable from those recorded in vivo near the primary BPs of PTs (Fig. 5c). This consistency is indeed remarkable, as we did not constrain, tune, or select the models to match these in vivo observations. Simultaneous recording and $Ca^{2+}$ imaging of whisker-evoked responses near the primary BP provided further in vivo evidence in support of our in silico predictions[18]. These experiments demonstrated that sensory-evoked $Ca^{2+}$ APs result generally in detectable $Ca^{2+}$ signals, whereas fast potentials do not (Supplementary Fig. S6). Consequently, we predict that whisker deflections evoke $Ca^{2+}$ signals in only a small minority of trials (3 ± 4%, $n = 360$ PT models). Our analysis of

in vivo $Ca^{2+}$ signals in apical trunks of PTs (3 ± 5%, $n = 229$ PTs) – imaged upon whisker deflections by Takahashi et al.[11], – supported this prediction (Fig. 5d). Taken together, consistent with in vivo recordings and $Ca^{2+}$ imaging[11,18,46,47], our simulations provided model consensus that sensory input evokes most frequently fast dendritic potentials, and in a small minority of trials $Ca^{2+}$ APs.

Sensory-evoked $Ca^{2+}$ APs occurred in simulation trials with all somatic response types (Fig. 5e). These in silico predictions are supported by in vivo experiments that combined somatic recordings with dendritic $Ca^{2+}$ imaging – i.e., the study observed $Ca^{2+}$ signals during responses with 1 AP, or with bursts of 2 or 3 APs[18]. Both in vivo and in silico results hence indicate that observing a $Ca^{2+}$ AP is generally insufficient to infer the response that sensory input will evoke at the soma (Fig. 5f). However, the in vivo data seem to suggest that the inverse may be possible – i.e., to infer the occurrence of a $Ca^{2+}$ AP from

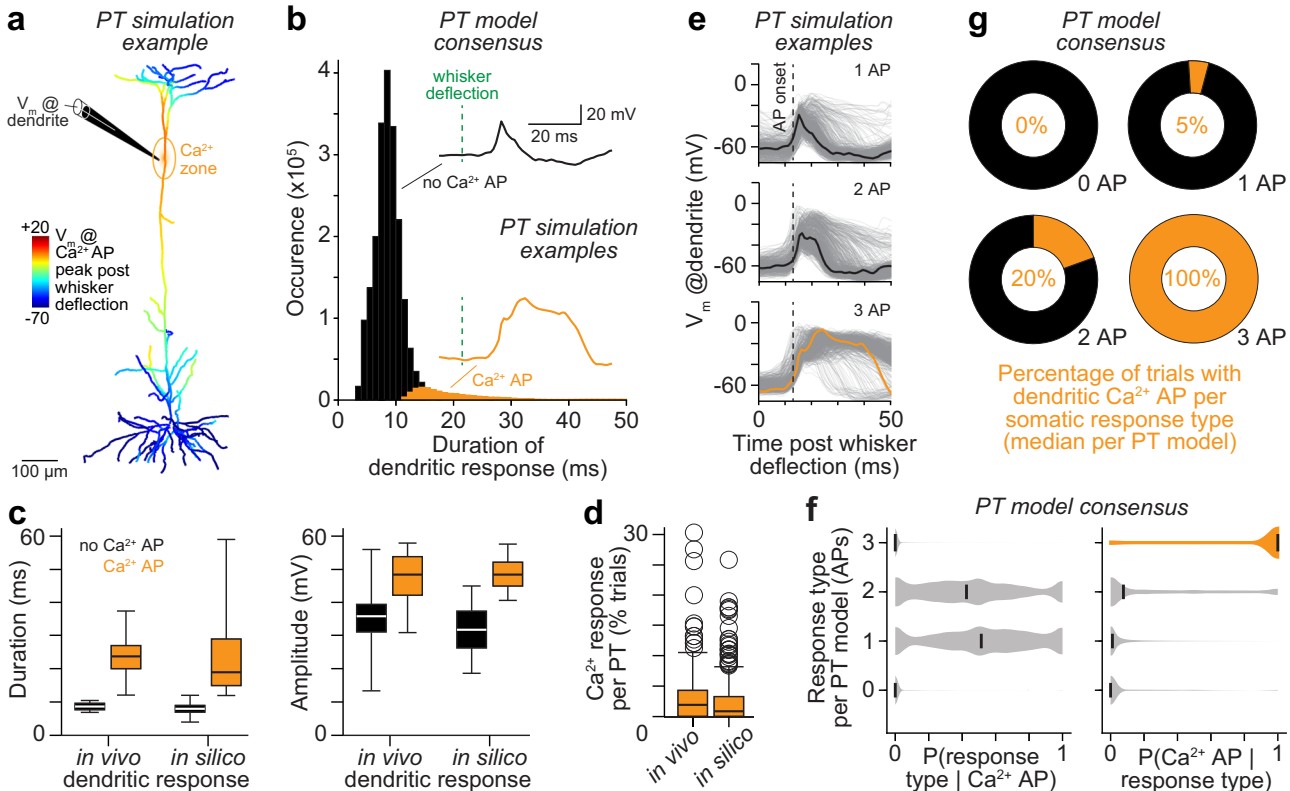

**Fig. 5 | In silico predicted dendritic activity during sensory-evoked cortical output. a** Example simulation for the PT model example in Fig. 3e shows membrane potential ($V_m$) during Ca²⁺ AP, which is predicted to be evoked by passive whisker deflection. **b** Histogram of the durations of sensory-evoked dendritic potentials. The vast majority of the dendritic responses are characterized by fast potentials (black). Dendritic Ca²⁺ APs occur only in a small minority of the trials (orange). **c** Durations (left) and amplitudes (right) of the in silico predicted dendritic responses ($n$ = 197,636 trials with Ca²⁺ APs, 1,048,576 trials without Ca²⁺ APs) versus those reported previously from whole-cell recordings near the primary BP of PTs in the barrel cortex for the same experimental condition (i.e., passive whisker deflections in anesthetized rats)[46]. Box plots represent medians and 25th to 75th percentiles. Whiskers extend to 1.5 times the interquartile range. **d** We quantified the occurrences of sensory-evoked Ca²⁺ signals that Takahashi et al. measured via Ca²⁺ imaging in apical dendrites of PTs in barrel cortex upon passive whisker deflections ($n$ = 229 PTs)[11], which matched the in silico predicted sensory-evoked

Ca²⁺ AP occurrences ($n$ = 360 PT models). Box plots represent medians and 25th to 75th percentiles. Whiskers extend to 1.5 times the interquartile range. **e** Membrane potentials at the Ca²⁺ domain for 1000 example simulations with sensory-evoked 1 AP responses (top), bursts of 2 APs (center) and bursts of 3 APs (bottom), respectively. The orange trace highlights exemplary sensory-evoked Ca²⁺ AP corresponding to panel (**a**). **f** Left: Probability across all PT models that somatic responses without APs occur (i.e., 0 APs), or with 1 AP, bursts of 2 APs or bursts of 3 APs occur, respectively, during trials when sensory-evoked Ca²⁺ APs occur. Right: Probability across all PT models that sensory-evoked Ca²⁺ APs occur during trials with 0 APs, or 1 AP responses or bursts of 2 APs or bursts of 3 APs, respectively. **g** Occurrences of Ca²⁺ APs differ substantially depending on the somatic response – i.e., virtually all trials with a burst of 3 APs, and across all PT models, showed a Ca²⁺ AP, whereas a vast majority of trials with 0, 1 or 2 AP responses showed no Ca²⁺ AP, but instead a fast dendritic depolarization (see also Supplementary Fig. S7). Source data for panels (**c**, **d** and **f**) are provided in the Source Data file.

the somatic response[18]. Ca²⁺ signals occurred reliably during bursts of 3 APs, whereas responses with 1 AP or with bursts of 2 APs also occurred in the absence of Ca²⁺ signals[18]. Our simulations supported this interpretation. Across PT models, virtually all trials with bursts of 3 APs had Ca²⁺ APs (Fig. 5g). In contrast, a vast majority of trials with somatic responses of 1 AP, or bursts of 2 APs had no Ca²⁺ APs, but instead fast potentials as their dendritic responses (Supplementary Fig. S7). Thus, supported by simultaneous in vivo recordings and Ca²⁺ imaging[18], our simulations provided model consensus that a burst of 3 APs is generally sufficient to infer the occurrence of a sensory-evoked Ca²⁺ AP (Fig. 5f). In essence, we predict that a burst of 3 APs following sensory stimulation requires interactions in the apical dendrites of PTs that generate Ca²⁺ APs, whereas the other response types do not.

Which interactions in apical dendrites generate Ca²⁺ APs during bursts of 3 APs? To address this question, we revisited our in silico manipulations and investigated their impact on apical dendrites. Depriving PT models from all indirect sensory input (manipulation 1) abolished all Ca²⁺ APs (Fig. 6a). The dendritic responses that remained had low amplitudes (11.0 ± 6.9 mV vs. 50.0 ± 4.7 mV during control) and fast onsets that preceded all sensory-evoked APs, on average by

1.8 ms. These fast, but weak, dendritic responses did not originate from bAPs, but instead from the direct sensory input to the apical trunk. Depriving PT models from direct sensory input (manipulation 2) also abolished Ca²⁺ APs (Fig. 6a), particularly when we reduced direct sensory input around the primary BP (manipulation 10, Fig. 6b). The dendritic responses that remained had low amplitudes (18.6 ± 8.5 mV), but were delayed, on average by 2.9 ms, and hence succeeded the first sensory-evoked APs. These manipulations provided model consensus that direct and indirect sensory input alone are each too weak to drive Ca²⁺ APs. Instead, direct sensory input to apical dendrites results in weak depolarization of the PTs' Ca²⁺ domain, which precedes their first sensory-evoked somatic AP. Thus, direct sensory input shifts the membrane potential of the apical trunk, and in particular of the Ca²⁺ domain around the primary BP, closer to the threshold for generating Ca²⁺ APs. Therefore, indirect sensory input streams that impinge subsequently onto the apical dendrites, and which would otherwise be too weak, can generate sensory-evoked Ca²⁺ APs. As shown above, bursts of 3 APs in sensory-evoked cortical output, but not responses of 1 AP or bursts of 2 APs, rely on a transition of fast, but weak, TC-driven dendritic responses into Ca²⁺ APs.

Additional in silico manipulations support our results that the specific and dense innervation of the apical trunk by TC synapses enables the generation of $Ca^{2+}$ APs in PTs, and hence the modulation of sensory-evoked cortical output with bursts of 3 APs. Prior to the direct sensory input, apical dendrites were generally more depolarized when PTs responded with bursts of 3 APs, on average by 1.4 mV (Wilcoxon rank-sum test: $p = 5.4^{-11}$). This increased depolarization even preceded the onset of the stimulus (Fig. 7a). To directly test the effect of prestimulus inputs on bursts of 3 APs, we reduced excitatory input to PTs during 20 ms preceding the direct sensory input (manipulation 11). Indeed, reduced ongoing input to apical dendrites, but not to basal dendrites, reduced the occurrences of sensory-evoked $Ca^{2+}$ APs (Fig. 7b, c) and of bursts of 3 APs (Fig. 7b/d). These manipulations indicate that the more depolarized apical dendrites are prior to direct sensory input (Fig. 8a), the closer will the direct sensory input shift the membrane potential of the $Ca^{2+}$ domain to the threshold (Fig. 8b), and hence the more effectively can indirect sensory input drive $Ca^{2+}$ APs (Fig. 8c). We term this mechanism for generating sensory-evoked $Ca^{2+}$ APs 'TC coupling', as it relies on the fast, but weak, local activation of the $Ca^{2+}$ domain by direct input from the thalamus, and because it enables PTs to couple information from multiple sensory-evoked and ongoing input streams. Bursts of 3 APs – but not bursts of 2 APs – in sensory-evoked cortical output provide a neurophysiological signature for TC coupling (Fig. 8d).

## Testing the in silico predicted origins of bursts in sensory-evoked cortical output

How can we test whether the in silico predicted TC coupling mechanism exists in vivo? An obvious strategy would be to repeat our in silico manipulations in vivo, while monitoring the membrane potential at the soma and dendrites of PTs. Such a direct demonstration of TC coupling remains an open challenge due to several technical limitations (see Discussion). Even so, we considered an alternative strategy to test TC coupling in vivo (Fig. 9a). In essence, TC coupling predicts that PTs, which respond with 1 AP to (direct and indirect) TC input, should respond with bursts to the exact same TC input, conditional on the additional inputs that they receive (e.g., from ongoing activity). Moreover, this transition should affect bursts of 2 and 3 APs differently. Thus, to provide indirect in vivo evidence in support of TC coupling, we investigated whether and how ongoing activity in the barrel cortex modulates TC-driven cortical output with bursts. This was possible because the virus we had injected into the VPM expressed the light-gated ion channel ChR2 in the presynaptic TC terminals. This enabled us to activate TC synapses with 10 ms light pulses while recording in vivo from the very same neurons that were the basis for our multi-scale simulations. Virtually all excitatory and inhibitory neurons that we recorded from in the barrel cortex (56/61) – including PTs (22/24) – responded to the light. In fact, irrespective of their type and laminar location, the neurons responded in almost all trials (trials per neuron with light-evoked APs, median: 97%), equally fast (AP onset per neuron, median/25th/75th: 5/4/7 ms), and showed only small variations between trials (variability of AP onset per neuron: 0.5/0.2/1.0 ms). Light-evoked responses of PTs (AP onset: 6/5/8 ms) hence originated from direct and indirect TC-driven input streams (Supplementary Fig. S8a), and these inputs were nearly identical across trials. Thus, our optogenetic experiments provided a fixed TC-driven input to PTs – a requirement for our strategy to test TC coupling in vivo.

We tested in silico how ongoing activity in the barrel cortex should modulate TC-driven PT output. For this purpose, we performed simulations in which we provided fixed TC-driven inputs and scaled the ongoing firing rates that we had recorded in the barrel cortex. Indeed, a diverse set of PT models responded with 1 AP, or bursts of 2 or 3 APs, conditional on how we scaled ongoing firing rates (Fig. 9b). Moreover, the simulations predicted four observations, which should characterize the relationship between ongoing activity and TC-driven

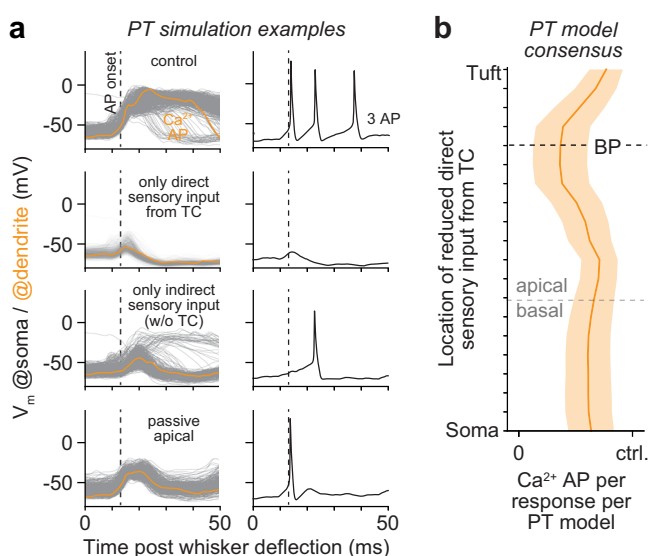

**Fig. 6 | Impact of in silico manipulations on sensory-evoked $Ca^{2+}$ APs.** **a** Simulation examples ($n = 1000$ as in Fig. 5b) show how in silico manipulations affect $Ca^{2+}$ APs (left) and APs during sensory-evoked bursts of 3 APs. From top to bottom: no manipulation (i.e., control), we deprived PTs from any indirect sensory input via TC-driven neurons in vS1 (i.e., responses are evoked solely by direct sensory input via the TC→PT pathway), we deprived PTs from any direct sensory input via the TC→PT pathway (i.e., responses are evoked solely indirect sensory input via TC-driven neurons in vS1), we deprived PTs from the active properties of their apical dendrites (i.e., PTs could not generate $Ca^{2+}$ APs). **b** Model consensus: sensory-evoked dendritic $Ca^{2+}$ APs during responses with bursts of 3 APs require direct sensory input to apical dendrites – in particular around the primary BP. Shadings represent the 25th to 75th percentiles, bold line represents the median. Source data for panel (**b**) are provided in the Source Data file.

PT output for our experimental condition (Fig. 9c, d): (1) for baseline ongoing activity, PTs should not respond with bursts of 3 APs; (2) small increases of ongoing activity (< 1.2 times the in vivo observed firing rates) should facilitate transitions almost exclusively to bursts of 2 APs; (3) larger increases should facilitate transitions to bursts of 2 and 3 APs; (4) once bursts of 3 APs occur more frequently than 1 APs (> 1.5 times the in vivo observed firing rates), occurrences of bursts of 2 APs should decline.

We tested in vivo whether ongoing activity in the barrel cortex modulates TC-driven PT output as we predicted in silico. For this purpose, we quantified the light-evoked PT responses ($n = 24$ PTs from $N = 17$ rats) conditional on ongoing activity in the barrel cortex (Fig. 9e), which we estimated as the LFP amplitude at the onset of each light stimulus compared to the baseline before the first light stimulus (Supplementary Fig. S8b). In trials where ongoing activity did not increase above baseline, PTs responded with 1 AP or with bursts of 2 APs, but bursts of 3 APs were generally absent. Small increases of ongoing activity (LFP increase < 1 mV) facilitated primarily the transition to bursts of 2 APs, while larger increases also facilitated the transition to bursts of 3 APs. Once bursts of 3 APs occurred more frequently than 1 APs (LFP increase > 2.5 mV), occurrences of bursts of 2 APs declined. These observations (Fig. 9f, g) agreed well with the predicted relationship between ongoing activity and TC-driven PT output (Fig. 9c, d). Indeed, we demonstrate in vivo that ongoing activity modulates TC-driven PT output with bursts, and that this modulation differs for bursts of 2 and 3 APs. Bursts of 2 APs can originate from TC-driven input alone. Bursts of 3 APs required additional input from ongoing cortical activity. Moreover, no other excitatory or inhibitory neurons in the barrel cortex increased bursts of 3 APs conditional on ongoing activity

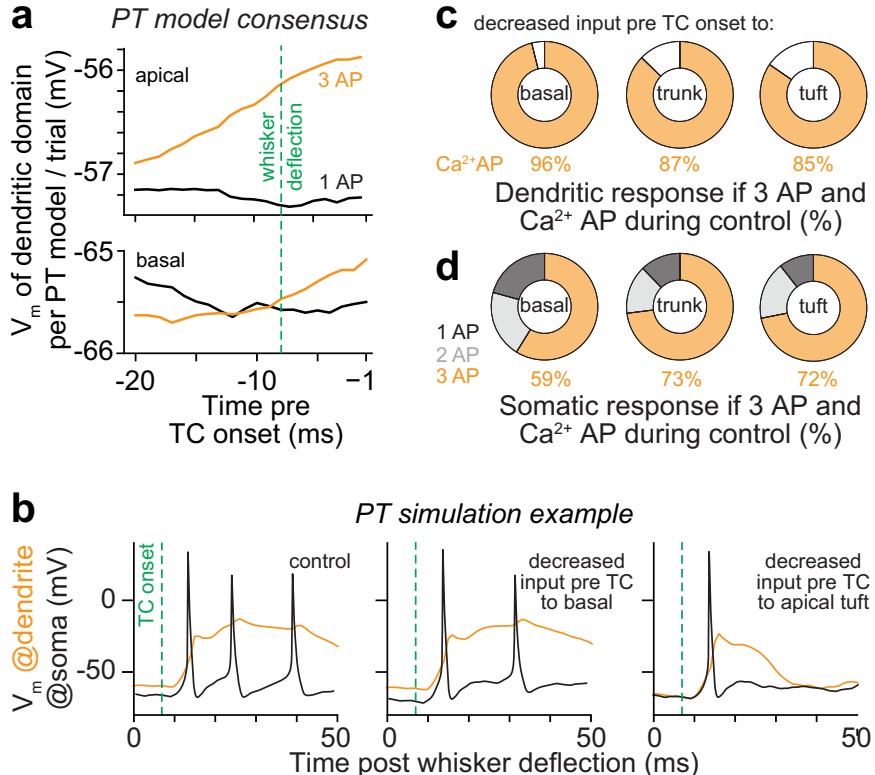

**Fig. 7 | In silico predicted impact of prestimulus inputs on sensory-evoked cortical output. a** Model consensus: membrane potentials in the apical dendrites (top), but not in the basal dendrites (bottom), are more depolarized before direct sensory input from the thalamus to PTs (TC onset) in trials with bursts of 3 APs. **b** Simulation examples show how in silico manipulations preceding the TC onset affect bursts of 3 APs. From left to right: no manipulation, we decreased input during 20 ms pre TC onset to basal or apical dendrites, respectively (**c**). Model consensus: sensory-evoked Ca²⁺ APs during responses with bursts of 3 APs are primarily reduced when we reduced input pre TC onset to the apical dendrites (both trunk or tuft), not basal dendrites. **d** Model consensus: bursts of 3 APs are reduced when we reduced input pre TC onset to both basal and apical dendrites. Source data for panel (**a**) are provided in the Source Data file.

(Supplementary Fig. S8c). Overall, these in vivo results support the in silico predicted TC coupling mechanism.

Do our predictions generalize to other experimental conditions? We quantified the activity of PTs ($n = 17$) in the barrel cortex of awake rats ($N = 15$) to address this question. Rats were not trained to perform tactile behavior. Instead, sensory responses were evoked by whisker touch of a pole during voluntary rhythmic whisker movements[15] – i.e., whisking (Fig. 10a). We chose this experimental condition, because such active touches were reported to lead to a ~3-fold increase of the sensory-evoked Ca²⁺ signals in apical dendrites of PTs, likely due to additional input to upper layers of the barrel cortex from whisker-related activity in motor cortex[47]. According to TC coupling, a 3-fold increase of sensory-evoked Ca²⁺ signals (i.e., putative Ca²⁺ APs[18]) should facilitate specifically the occurrences of bursts of 3 APs. This was indeed the case. We found that active touches evoked the same three response types (Fig. 10b) that we had observed for whisker deflections (i.e., passive touches). In fact, PTs responded to active touches with 1 AP ($p = 0.48$) or bursts of 2 APs ($p = 0.45$) as frequently as they did to passive touches, whereas bursts of 3 APs occurred on average 3.7 times more frequently upon active touches ($p = 0.02$). Thus, active and passive touches evoked virtually identical responses across PTs (Mann-Whitney U test of response probability, 1-sided: $p = 0.98$), with the exception of bursts of 3 APs ($p = 0.017$) (Fig. 10c). Moreover, compared to periods of quiescence and whisking (Fig. 10d), occurrences of bursts of 3 APs increased upon active touch (1-way ANOVA with multiple comparison: $p = 0.02$), whereas occurrences of 1 AP ($p = 0.65$) or bursts of 2 APs ($p = 0.08$) did not. This result supports observations by Takahashi et al., which showed that active behavioral engagement also

increases occurrences of sensory-evoked Ca²⁺ signal in PTs[11]. Taken together, experimental conditions with increased occurrences of sensory-evoked Ca²⁺ signals in apical dendrites[11,12,47] also increase specifically and to similar degrees occurrences of sensory-evoked bursts of 3 APs. These results provide further in vivo evidence in support of TC coupling. In any event, they demonstrate that specifically bursts of 3 APs in sensory-evoked cortical output occur conditional on stimulus type and behavioral state.

## Discussion

It is well known that PTs could utilize dendritic Ca²⁺ APs to perform *coincidence detection* – i.e., to multiplex inputs that impinge near simultaneously onto the basal and apical dendrites into bursts of somatic APs[16]. Currently, back-propagating APs (bAPs) are considered as the major mechanism that enables coincidence detection[16]. Our results suggest two major alterations of this perspective. First, we find that axons from the primary thalamus target specifically and most densely the dendritic Ca²⁺ domain of PTs. We show that direct sensory input from the thalamus hence provides a weak depolarization to the Ca²⁺ domain that does not rely on bAPs. Second, in contrast to bAPs, we find that the weak depolarization of the Ca²⁺ domain by the thalamus precedes the first sensory-evoked somatic AP. We show that this fast activation of the Ca²⁺ domain widens the timing for coincidence detection to prestimulus periods – i.e., in addition to apical inputs that arrive after a sensory-evoked AP[16], also those that arrive before it can contribute to the generation of sensory-evoked Ca²⁺ APs, and hence to the modulation of PT output with bursts. Taken together, our results reveal a novel mechanism by which PTs could utilize dendritic Ca²⁺ APs

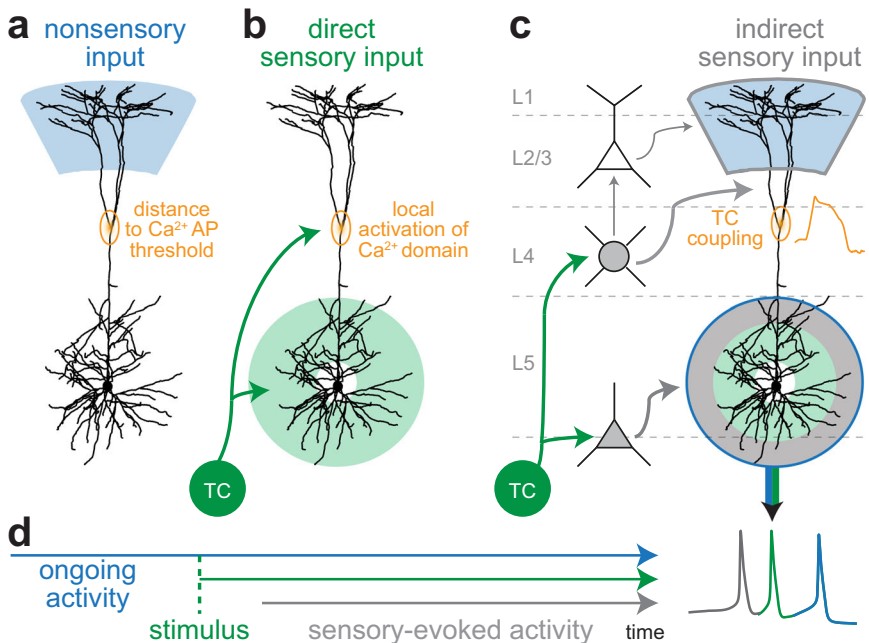

**Fig. 8 | Schematic illustration of TC coupling mechanism. a** Input to the apical dendrites of PTs (blue) that precedes the onset of direct sensory input from the thalamus determines how far the dendritic membrane potential is away from the Ca²⁺ AP threshold. **b** The more depolarized the Ca²⁺ domain is prior to the stimulus, the closer will the direct sensory input from the thalamus (green), which specifically targets this domain, shift it to the Ca²⁺ AP threshold. **c** This fast, but weak, TC-driven activation of the Ca²⁺ domain enables indirect sensory input streams (gray), which would otherwise be too weak, to drive sensory-evoked Ca²⁺ APs (e.g., TC→L4SPs→PT). Bursts of 3 APs rely on this transition of the TC-driven dendritic responses into Ca²⁺ APs. **d** Indirect sensory input streams (i.e., TC→L6CC→PT[32,35]) are sufficient to evoke 1 AP responses (gray). Multiplexing of direct and indirect sensory inputs in basal dendrites is sufficient to modulate 1 AP responses with bursts of 2 APs (green). However, additional inputs to apical dendrites (sensory-evoked and/or nonsensory) are required to modulate sensory-evoked cortical output with bursts of 3 APs (blue). This multiplexing of basal with apical inputs into bursts of 3 APs arises from the specific innervation of the Ca²⁺ domain by the TC→PT pathway. We hence term this mechanism 'TC coupling'.

to perform coincidence detection, which we term *TC coupling*, and which enables pre- and poststimulus apical inputs to modulate the first sensory-evoked responses that leave the cortex with bursts. Our findings support the interpretation by Takahashi et al. that sensory-evoked Ca²⁺ signals in apical dendrites of PTs do not result from bAPs, but instead from a local activation of the Ca²⁺ domain[12] – i.e., via TC coupling.

We will discuss the implications of TC coupling for downstream sensory processing below. However, before proceeding, we will review some of the technical features of the approach we employed. Our findings result from in vivo recordings of morphologically identified PTs, which we combined with reconstructions of TC synapses and multi-scale simulations. Our multi-disciplinary approach enabled us to investigate in silico how empirically observed TC synapse distributions along the dendrites of in vivo recorded PTs could, in principle, contribute to their sensory-evoked responses. We did these investigations for a comprehensive set of PT models, which captured the empirically observed ranges of parameters from subcellular to network scales. These PT models reached consensus that a fast, but weak, depolarization of the dendritic Ca²⁺ domain by direct sensory input from the thalamus, and hence TC coupling, is an origin of bursts in sensory-evoked cortical output.

A clear demonstration of our simulations is the finding that sensory-evoked bursts of 3 APs have different origins than bursts of 2 APs. Both burst types require interactions of direct with indirect sensory inputs in the basal dendrites of PTs, but bursts of 3 APs require additional interactions of direct with indirect sensory inputs in the apical dendrites, as well as the active properties of the dendritic Ca²⁺ domain. In essence, the different mechanistic origins of the two types of sensory-evoked bursts enabled us to test our in silico predictions for TC coupling in vivo.

Three in vivo experiments, in both anesthetized and awake rats, support our in silico results about the different origins for sensory-evoked bursts of 2 and 3 APs. We show that specifically bursts of 3 APs occur conditional on stimulus type (active vs. passive whisker touch) and behavioral state (anesthetized vs. awake; quiescent or whisking vs. active touch). Moreover, we demonstrated via optogenetic activation of TC synapses that a fixed TC-driven input drives bursts conditional on ongoing activity in the barrel cortex, and that ongoing activity differentially affects bursts of 2 and 3 APs. In fact, the optogenetic manipulations revealed relationships between ongoing activity and TC-driven PT output that were highly consistent with those predicted in silico. Noticeably, the manipulations provided direct in vivo evidence for our in silico prediction that TC-driven input is sufficient to evoke bursts of 2 APs, whereas bursts of 3 APs require additional input – e.g., from ongoing activity. Our in vivo experiments hence support our conclusion that TC coupling underlies the differences between the two types of bursts in sensory-evoked cortical output.

We emphasize that a direct in vivo demonstration of TC coupling remains an open challenge. This would require the ability to deprive PTs in vivo from direct sensory input specifically to the Ca²⁺ domain, without affecting any other direct and TC-driven indirect sensory input streams. Until such manipulations become feasible, in vivo demonstrations of TC coupling will remain limited to indirect tests, as we did here. Thus, despite in silico consensus and its support by several lines of in vivo evidence, we cannot rule out the possibility that other mechanisms account, at least in part, for some of the bursts that we recorded in vivo. We addressed this possibility by investigating two scenarios. First, synaptic plasticity in the TC→PT pathway could strengthen the direct TC input to the Ca²⁺ domain in response to repetitive optogenetic stimulation[48], and thereby facilitate bursts. Second, our optogenetic stimulations could indirectly activate

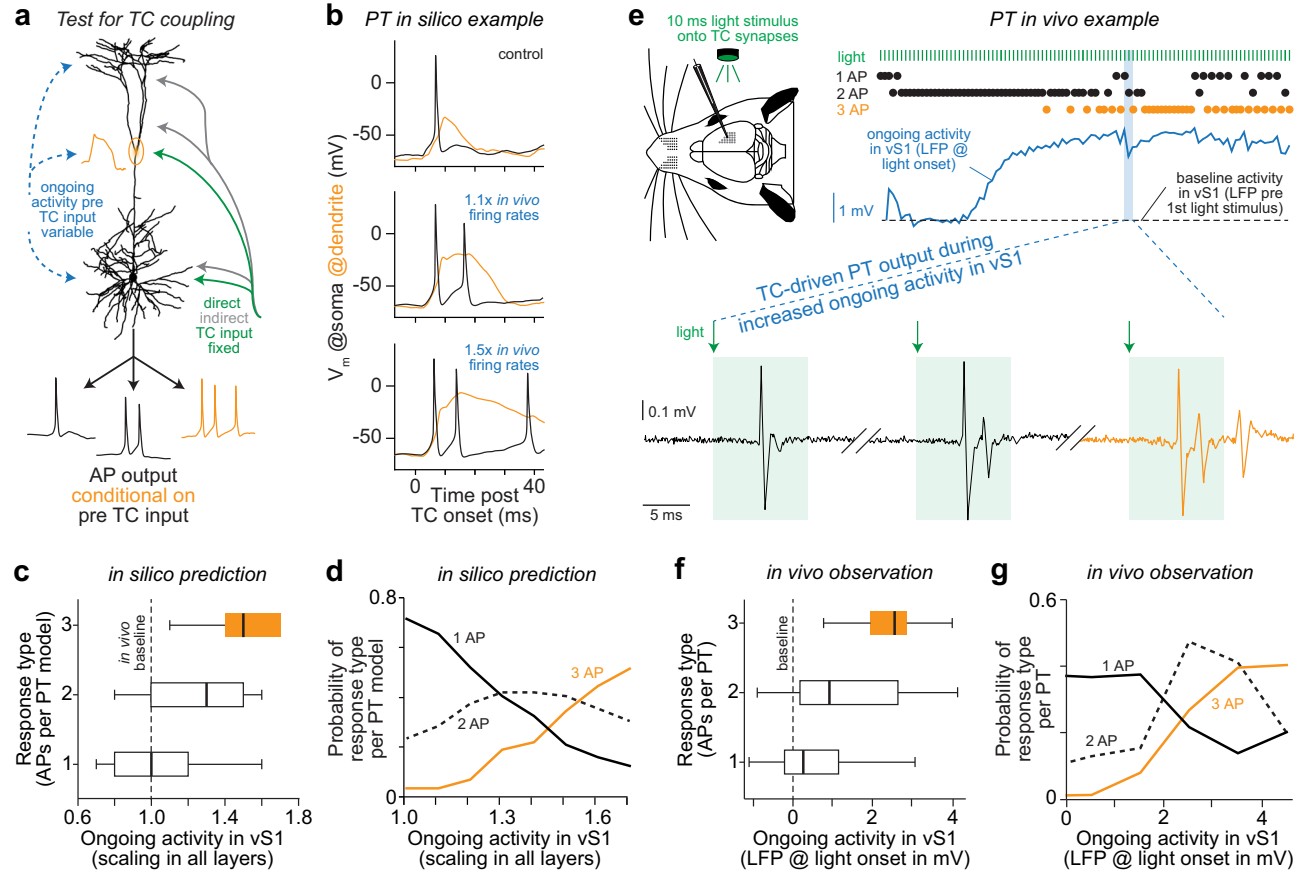

**Fig. 9 | Testing the predicted origins of bursts in sensory-evoked cortical output. a** Schematic illustration of the proposed strategy. In response to the same direct and indirect input from the thalamus, PTs should transition from 1 AP responses into bursts conditional on their additional inputs, e.g., from ongoing activity, and these transitions should be different for bursts of 2 and 3 APs. **b** Simulation examples show how in silico manipulations of ongoing activity affect responses with 1 AP. From top to bottom: no manipulation, we scaled ongoing firing rates beyond those that we had recorded in the barrel cortex in control conditions by a factor of 1.1 and 1.5 for excitatory types in all layers. **c, d** In silico predicted relationships between scaling of ongoing activity and TC-driven PT output for our experimental in vivo condition. Box plots represent medians and 25th to 75th percentiles. Whiskers extend to 1.5 times the interquartile range (3237, 2108, 1193 in silico responses with 1, 2, 3 APs). **e** Left: schematic of our optogenetic experiments in anesthetized rats. The schematic of the rat was adapted and modified from

"Diamond ME, von Heimendahl M, Knutsen PM, Kleinfeld D, Ahissar E. 'Where' and 'what' in the whisker sensorimotor system. Nat Rev Neurosci 9, page 602, 2008, Springer Nature"[68]. Reproduced with permission from Springer Nature. Right: example recording for one of the PTs whose TC input along the dendrites is shown in Fig. 1c (most right), and whose in vivo and in silico responses are shown in Fig. 2a, b and Figs. 3–6. Bottom: the traces show somatic APs recorded during three example trials in which we activated TC synapses by a 10 ms light pulse (green shading). We estimated ongoing activity as the LFP amplitude in the barrel cortex at the onset of each light stimulus (blue) compared to baseline before the first light stimulus (dashed line). **f, g** In vivo relationships between ongoing activity and TC-driven PT output (24 PTs from 17 rats) analogous to the in silico predictions in panels (**d, e**). Box plots represent medians and 25th to 75th percentiles. Whiskers extend to 1.5 times the interquartile range (2025, 744, 197 in vivo responses with 1, 2, 3 APs). Source data for panels (**c, d, f** and **g**) are provided in the Source Data file.

additional inputs that impinge onto PTs from other long-range pathways in the thalamus[49] and cortex[47], and thereby facilitate bursts. We performed additional simulations that mimic these scenarios. Synaptic plasticity, as observed in vitro (Supplementary Fig. S9), as well as additional inputs to apical dendrites (Supplementary Fig. S10), could indeed facilitate the generation of Ca²⁺ APs. However, while bursts of 3 APs increased in both scenarios, neither an increase due to TC plasticity (Supplementary Fig. S9f) nor due to additional inputs (Supplementary Fig. S10c) resulted in responses consistent with our in vivo observations. Overall, we conclude that TC coupling underlies the burst responses of 3 APs that we observed in anesthetized and awake rats.

**Implications of TC coupling for downstream sensory processing**
The POm thalamus is one of the primary downstream targets for PTs in the barrel cortex[42]. This corticothalamic pathway forms giant terminals with strong synapses[50]. Sensory-evoked cortical output thereby reliably activates TC neurons in the POm, which then provide feedback to

the barrel cortex[25,28,51] and to other cortical areas, including frontal cortices[52]. Due to strong depression of the PT→POm synapse[50], this pathway filters out bursts that arrive in isolation from single PTs[5,9]. In turn, bursts that arrive from multiple PTs simultaneously can overcome the effects of depression and thereby facilitate burst firing in the POm neurons[6,8]. Such bursts in the POm feedback enhance sensory processing in the barrel cortex[7], and likely, in other cortical areas that receive POm input[49]. Thus, synchronous timing of bursts is critical for the ability of PTs to drive enhanced interactions between thalamus and cortex. We showed that direct sensory input from the VPM thalamus sets the timing of bursts. The first sensory-evoked APs hence occur simultaneously across trials and across PTs, irrespective of whether the responses are 1 AP or bursts of 2 or 3 APs. Thus, by increasing the number of PTs in the barrel cortex that transmit simultaneously bursts to the POm, TC coupling is ideal to drive enhanced interactions between thalamus and cortex during sensory processing.

Output from the barrel cortex to POm is critical for the ability of rodents to perceive sensory input from the whiskers[10]. Moreover,

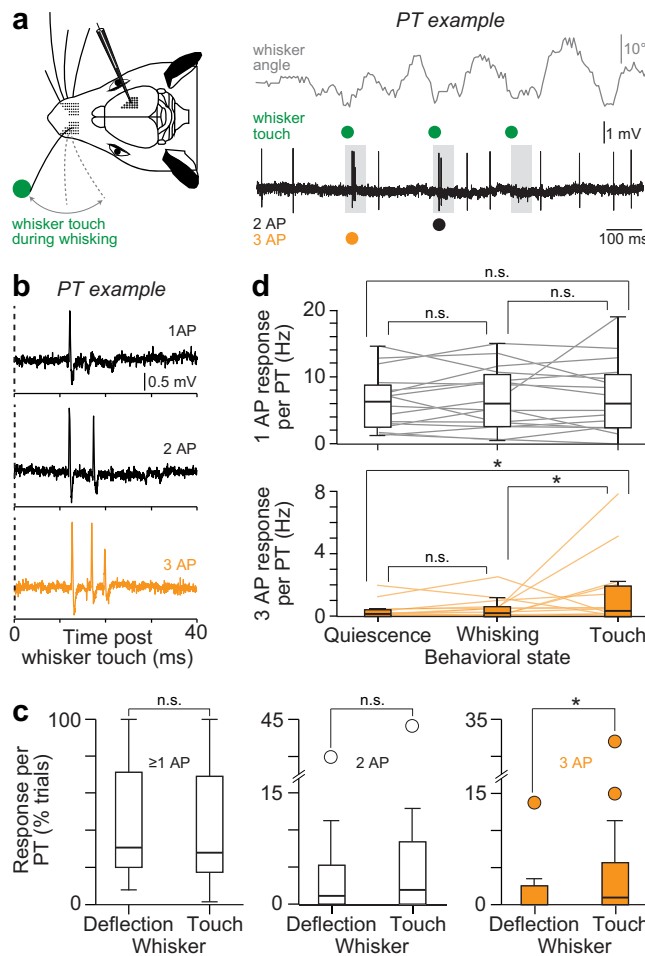

**Fig. 10 | Bursts in sensory-evoked cortical output of awake animals. a** Left: schematic of our experiments in vS1 of awake rats[15]. The schematic of the rat was adapted and modified from "Diamond ME, von Heimendahl M, Knutsen PM, Kleinfeld D, Ahissar E. 'Where' and 'what' in the whisker sensorimotor system. Nat Rev Neurosci 9, page 602, 2008, Springer Nature"[68]. Reproduced with permission from Springer Nature. Right: example recording for one PT. The trace shows APs recorded during three example trials in which the rat touched a pole with a single whisker during periods of voluntary rhythmic whisker movements (whisking). The gray boxes denote the 50 ms time window for which we analyzed sensory-evoked responses. **b** Example trials of the PT from panel a show that active touch evokes the same three types of sensory responses that we observed for passive whisker stimuli in Fig. 2. **c** Comparison of responses (≥ 1AP) and occurrences of bursts of 2 and 3 APs between passive whisker deflections (25 PTs) in anesthetized rats ($N = 18$) and active whisker touches (17 PTs) in awake rats ($N = 15$). Box plots represent medians and 25th to 75th percentiles. Whiskers extend to 1.5 times the interquartile range. The asterisk represents the Mann-Whitney U test, 1-sided: $p = 0.017$. **d** Top: 1 APs during periods of quiescence, whisking and active touch ($n = 1763$ trials, $N = 16$ PTs). Bottom: same as in the top panel, but for bursts of 3 APs. Box plots represent medians and 25th to 75th percentiles. Whiskers extend to 1.5 times the interquartile range. The asterisks represent 1-way ANOVA tests with PostHoc Tukey multiple comparison correction for quiescence vs. touch $p = 0.023$ and whisking vs. touch $p = 0.041$. Source data for panels (**c**, **d**) are provided in the Source Data file.

sensory-evoked Ca²⁺ signals in the apical dendrites of PTs in the barrel cortex correlate with the threshold for perceiving whisker stimuli[11,12]. It is tempting to speculate that TC coupling underlies, at least in part, these observations. Specifically, TC coupling enables ongoing activity to facilitate sensory-evoked Ca²⁺ APs, and hence to increase bursts of 3 APs in descending cortical output. Therefore, nonsensory information that impinges onto the apical dendrites before (and during) a stimulus

can influence whether and how strongly a stimulus drives interactions with subcortical regions. Nonsensory information may thereby influence whether a stimulus is perceived or not. Observations in humans support this speculation. Visual stimuli are only perceived if they evoke responses in the primary visual cortex that ignite enhanced interactions across large-scale prefrontal-parietal cortical networks[53]. By enabling nonsensory information to modulate the first sensory responses that leave the cerebral cortex with bursts, TC coupling may hence be critical for igniting the cascade of interactions between cortical and subcortical regions that transform sensory input into perception.

Overall, it is remarkable that long-range axons are capable of targeting a specific dendritic domain in a specific cell type, even if the target location varies from cell to cell. In essence, and supported by electron-microscopic studies[30,54], our findings indicate that the dendritic domain around the primary BP is not only distinguished by its biophysical properties, but also by its specific synaptic innervation. Other pathways may hence also target specifically the dendritic Ca²⁺ domain of PTs. Thus, TC coupling may represent a general circuit theme that enables large pyramidal neurons to multiplex anatomically segregated input streams into different types of burst output.

## Methods

### Virus injection

Male Wistar rats were provided by Charles River Laboratories. All experiments were carried out after evaluation by local German authorities, and in accordance with the animal welfare guidelines of the Max Planck Society. Boutons along TC axons were virus-labeled as described previously[32]. Briefly, rats aged 22–25 days (P22–25) were anesthetized with isoflurane supplemented by Caprofen (5 mg/ kg) and Buprenorphine SR (1 mg/kg) as analgesia, then placed into a stereotaxic frame (Kopf Instruments, model 1900), and provided with a continuous flow of isoflurane/O2 gas. Body temperature was maintained at 37 °C by a heating pad. A small craniotomy was made above the left hemisphere 2.85 mm posterior to bregma and 3.2 mm lateral from the midline. The head of the rat was leveled with a precision of 1 μm in both the medial-lateral and anterior-posterior planes using an electronic leveling device (Sigmann Electronics, Hüffenhardt, Germany) mounted on the stereotaxic frame. An injecting pipette containing an adeno-associated virus[55] – rAAV2/1-Syn-hChR2(H134R)-mCherry (titer: $1 × 10^{12}$ gc ml⁻¹) – was lowered into VPM thalamus (5.05 mm from pia). Martin Schwarz (University of Bonn, Germany) provided the virus. 50–70 nL of virus were injected by a 30cc syringe coupled to a calibrated glass injection capillary.

### Electrophysiology in anesthetized animals

All experiments were carried out after evaluation by the local German authorities, and in accordance with the animal welfare guidelines of the Max Planck Society. Non-AAV-injected (P28–48) and AAV-injected rats (after a 16–21 day incubation period) were anesthetized with urethane (1.8 g/kg body weight) by intraperitoneal injection. The depth of anesthesia was assessed by monitoring pinch withdrawal, eyelid reflexes, and vibrissae movements. Body temperature was maintained at 37 °C by a heating pad. Cell-attached recording and Biocytin labeling was performed as described previously[56]. Briefly, a small craniotomy was made above the left hemisphere 2.5 mm posterior and 5.5 mm lateral to the bregma (for recordings in vS1), or 2.9-3.5 mm posterior to the bregma, 2.4-3.4 mm lateral from the midline, and at 5-6 mm depth from the pia for recordings in VPM thalamus. APs were recorded using an extracellular loose patch (ELC-01X, npi electronic GmbH) or an Axoclamp 2B amplifier (Axon instruments, Union City, CA, USA), digitized via a CED power1401 data acquisition board (CED, Cambridge Electronic Design, Cambridge, UK), and low-pass filtered (300 Hz) to measure the LFP. APs and LFPs were recorded before and during 20-30 trials of caudal multi-whisker deflections by a

700 ms airpuff (10 PSI), delivered through a 1 mm inner diameter plastic tube from a distance of 8–10 cm from the whisker pad[42]. ***Note:*** The air reached the whiskers 15 ms after we triggered the puff. Thus, stimulus onset in the raw data is 15 ms later than in the plotted data. Stimulation was repeated at 2.5 sec intervals. We assigned trials to the response types depending on their activity within the first 50 ms after stimulus onset: bursts were defined as 3 APs occurring within 30 ms, or as 2 APs within 10 ms. Trials that fulfilled both criteria were assigned as bursts of 3 APs. We used the same criteria for classifying optogenetic responses, active touch responses, and the simulations. ***Note:*** when the analysis window is extended beyond the first 50 ms, a small subset of bursts of 3 APs showed in silico and in vivo a 4th or even a 5th somatic AP (i.e., bursts of 3 APs likely generalize to bursts of ≥3 APs). We determined the latencies of TC onset at different cortical depths in vS1 by detecting the first depolarization of the LFP after stimulation. In AAV-injected rats, optical stimulation of TC terminals was provided by a 400 μm diameter optical fiber (ThorLabs #RJPSF2) coupled to a 470 nm LED source (ThorLabs M470F3) and powered by an LED driver (ThorLabs #DC2200). A 10 ms pulse of light generated 1 mW output power at the end of the optical fiber, as measured by a laser power meter (ThorLabs #PM100A) coupled to a photodiode (ThorLabs #S121C). We positioned the optical fiber 1–2 mm above the cortical surface via a 3-axis motorized micromanipulator (Luigs and Neuman), so that the light beam resulted in a 1–2 mm disc of light above the recording site in vS1. Control of the LED driver was implemented with Spike2 software (CED, Cambridge, UK). APs and LFPs were recorded during 20–100 trials of 10 ms light pulses, at intervals of 2.5, 0.5 and 0.1 seconds. Physiology did not differ between PTs with ($n = 10$) and without ($n = 15$) reconstructed TC synapses: ongoing firing rates (2-sided $t$ test unpaired $p = 0.2$), response probabilities to whisker stimuli ($p = 0.8$), latencies of these responses ($p = 0.2$), occurrences of 2 or 3 AP bursts in response to whisker stimuli ($p = 0.33$ or $0.98$), response probabilities to light stimuli ($p = 0.6$), latencies of light-evoked responses ($p = 0.5$), occurrences of 2 or 3 AP bursts in response to light stimuli ($p = 0.37$ or $0.34$). After the electrophysiological measurements, neurons were filled with Biocytin. After 1–2 h for biocytin diffusion, animals were transcardially perfused with 0.1 M phosphate buffer (PB) followed by 4% paraformaldehyde (PFA). Brains were removed and post-fixed with 4% PFA for 8–12 h, transferred to 0.1 M PB and stored at 4 °C.

### Electrophysiology in awake animals

All experiments were carried out in accordance with the animal welfare guidelines of the VU Amsterdam, the Netherlands. As described previously[15], male Wistar rats (P39 ± 4, provided by Charles River Laboratories) were positioned in the recording setup using a head-post. During surgical preparation, rats were anesthetized using 1.6% isoflurane in 0.4 l/h O2 + 0.7 l/h NO2 and the depth of anesthesia was assured by the absence of foot and eyelid reflexes. In addition, post-operative analgesia (buprenorphine, 0.1–0.5 mg/kg) was given. Body temperature was maintained at 37 °C with a heating pad. In the week prior to surgery, rats were handled daily to accustom them to the experimenter and housed in pairs in enriched cages. In the week after surgical preparation, rats were head-fixed twice per day for 2–3 days in preparation of the recording session. Rats quickly adjusted to the head-fixation period, allowing stable recording configurations without the need for body restraint. On the recording day, rats were anaesthetized with isoflurane (1.25% in 0.4 l/h O2 + 0.7 l/h NO2). After turning off NO2, we used intrinsic optical imaging (IOI) and passive single whisker stimulations to estimate the location of the D2 barrel column. We inserted pipettes for cell-attached recordings and Biocytin labeling at the hence determined D2 location, and repeated the whisker receptive field mappings for the single neurons to confirm the IOI results. Afterwards, all whiskers were clipped to 5 mm, except the principal (i.e., D2) or the single surround whisker that elicited the

maximal response (i.e., AP probability). After terminating the anesthesia, rats woke up within several minutes. APs during active object touches were quantified only after rats were fully awake, monitored by body posture and exploratory whisking. Active touch resulted from whisker self-motion monitored with high-speed videography (375 s continuously at 200 frames/sec, MotionScope M3 camera, IDT Europe, Belgium). A pole was positioned 2 cm lateral from the whisker pad and anterior relative to the whisker set point (obtained during quiescent episodes). This ensured that touches were the consequence of whisker protraction. We detected touches manually frame-by-frame based on three criteria: (1) no pixels between the whisker and the pole, (2) whisker protraction was blocked by pole contact, and (3) pole contact changed whisker curvature. After the recordings, we labeled the neurons with Biocytin to reveal *post hoc* their anatomical location relative to the barrel field. Only PTs whose anatomical location matched with the spared whisker were included in this study. Finally, rats were deeply anaesthetized with urethane (> 2.0 g/kg) and perfused with 0.9% NaCl, then 4% paraformaldehyde (PFA). Brains were post-fixed in 4% PFA overnight at 4 °C and transferred to 0.9% NaCl.

### Histology

For recordings in vS1 of non-AAV injected and awake rats, 22–25 consecutive 100 μm thick vibratome sections were cut tangentially to vS1 (45° angle), ranging from the pial surface to the white matter, or coronally (for recordings in VPM thalamus). All sections were treated with avidin-biotin (ABC) solution, and subsequently neurons were identified using the chromogen 3,3′-diaminobenzidine tetrahydrochloride (DAB). We mounted all sections on glass slides, embedded with Mowiol and enclosed with a cover slip. For recordings in vS1 of AAV injected rats, we cut the recorded cortical hemisphere into 45–48 consecutive 50 μm thick tangential vibratome sections from the pial surface to the white matter. The remaining brain tissue was embedded in 10% gelatin (Sigma Aldrich #G2500) and cut coronally into consecutive 100 μm thick sections to identify the injection site. Sections were washed 3 times in 0.1 M PB and treated with streptavidin conjugated to AlexaFluor488 (5 μg/ml) (Molecular Probes #S11223) in 0.1 M PB containing 0.3% Triton X-100 (TX) (Sigma Aldrich #9002-93-1), 400 μl per section for 3–5 h at room temperature in order to visualize Biocytin-labeled neurons. To enhance fluorescence expressed by the virus and to label TC synapses, we double-immunolabeled slices with anti-mCherry antibody and anti-VGlut2 antibody. Sections were permeabilized and blocked in 0.5% Triton x-100 (TX, Sigma Aldrich #9002-93-1) in 100 mM PB containing 4% normal goat serum (NGS, Jackson ImmunoResearch Laboratories #005-000-121) for 2 h at room temperature. The primary antibodies were diluted 1:500 (Rabbit anti-mCherry, Invitrogen #PA5-34974, Invitrogen #M11217 and mouse anti-VGlut2 antibody, Synaptic Systems #135421) in PB containing 1% NGS for 48 h at 4 °C. Secondary antibodies were diluted 1:500 (goat anti-Rabbit IgG Alexa-647 H + L Invitrogen #A21245 and goat anti-Mouse IgG Alexa-405 H + L Invitrogen #A31553), and were incubated for 2-3 h at room temperature in PB containing 3% NGS and 0.3% TX. All sections were mounted on glass slides, embedded with SlowFade Gold (Invitrogen #S36936) and enclosed with a coverslip.

### Morphological reconstructions

Neuronal structures were extracted from image stacks using a previously reported automated tracing software[57]. For reconstruction fluorescently labeled neurons and locating of AAV-labeled TC synapses, images were acquired using a confocal laser scanning system (SP5; Leica Microsystems). 3D image stacks of up to 2.5 mm × 2.5 mm × 0.05 mm were acquired at 0.092 × 0.092 × 0.5 μm per voxel (63 x, NA 1.3). Image stacks were acquired for each of the 45–48 consecutive tangential brain slices that range from the pial surface to the white matter. Manual proof-editing of individual sections and automated alignment across sections were done by custom-designed software[58].

Pia, barrel and white matter outlines were manually drawn on low-resolution images (4 x dry objective). Using these anatomical reference structures, all reconstructed morphologies were registered to a standardized reference frame of rat vS1[59]. The distance from the pial surface to the soma, and 20 morphological features that have previously been shown to separate between excitatory types in rat vS1[26] were calculated for each registered morphology (Supplementary Fig. S2b). For identification of putative TC synapses, Biocytin-labeled dendrites and AAV-labeled axons were imaged simultaneously using the confocal system as described above: Biocytin Alexa-488 (excited at 488 nm, emission detection range 495–550 nm), AAV Alexa-647 (excited at 633 nm, emission detection range 650–785 nm). The dual-channel image stacks were loaded into Amira software (Thermo Scientific), where we placed landmarks manually on dendritic spines if the spine head overlapped with a TC axonal bouton (i.e., the landmarks represent a putative TC synapse). The shortest distance of each landmark to the dendrite was determined, and the path length distance was calculated from that location to the soma. For validation of putative TC synapses, image stacks were acquired via super-resolution microscopy (SP8 LIGHTNING; Leica Microsystems) via a glycerol/oil immersion objective (HCX PL APO 63x, NA 1.3), a tandem scanning system (8 kHz resonance scanning speed), and spectral detectors with hybrid technology (GaAsP photocathode; 8x line average): VGlut2 Alexa-405 (excited at 405 nm, emission detection range: 410-480 nm), Biocytin Alexa-488 (excited at 488 nm, emission detection range 495-550 nm), AAV Alexa-647 (excited at 633 nm, emission detection range 650-785 nm). Triple-channel images were acquired at $29.5 \times 29.5 \times 130$ nm per voxel. Image stacks were visualized in Amira, and manually inspected for VGlut2 at contact sites between spines and AAV-infected TC boutons within single optical sections (range of VGlut2-positive contacts: 68-100%, $n = 349$ contacts across 26 basal and apical trunk dendrites of 8 PTs). We quantified TC synapse distributions along the dendrites of PTs as follows: First, we grouped TC synapses into those on apical trunks and tufts vs. those on basal and apical oblique dendrites. Second, for each group, we counted TC synapses per soma distance bin of 100 μm and divided these numbers by the dendritic length of the respective bin. Third, we set the locations of the soma, BP and most distal tuft for each PT to $x = 0$, 1 and 2, respectively, and linearly transformed the soma distance of the bins accordingly (i.e., bins between the soma and BP were scaled from 0 to 1, bins between the BP and distal tuft were scaled from 1 to 2). Finally, we computed the mean and STD of the TC synapse densities in this normalized distance space, as shown for the two groups in Fig. 1d, which allowed us to compare TC synapse densities by the relative soma-BP-tuft locations, even though the absolute soma-BP-tuft locations differed between PTs.

## Multi-compartmental models

We selected three morphologies from the in vivo recorded PTs with reconstructed TC synapses that capture the range of morphological variability (i.e., the PTs with the deepest and most superficial BPs, and one PT from the bulk of the BP distribution). Multi-compartmental models were generated for these dendrite morphologies as described previously[32,44]. Briefly, a simplified axon morphology was attached to the soma of the PT morphology[60]. The axon consisted of an axon hillock with a diameter tapering from 3 μm to 1.75 μm over a length of 20 μm, an axon initial segment of 30 μm length and diameter tapering from 1.75 μm to 1 μm diameter, and 1 mm of myelinated axon (diameter of 1 μm). Next, a multi-objective evolutionary algorithm was used to find parameters for the passive leak conductance and densities of Hodgkin-Huxley type ion channels on soma, basal dendrite, apical dendrite and axon initial segment, such that the neuron model is able to reproduce characteristic electrophysiological responses to somatic and dendritic current injections of PTs within the experimentally observed variability, including bAPs, $Ca^{2+}$ APs, and AP responses to

prolonged somatic current injections[44]. We augmented the original biophysical model of PTs[32,44] with two parameters: according to a previous report[61], the density of the fast non-inactivating potassium channels (Kv3.1) was allowed to linearly decrease with soma distance until it reaches a minimum density (i.e., the slope and minimum density are two additional parameters). The diameter of the apical dendrites was optimized by a scaling factor between 0.3 and 3. We incorporated the IBEA algorithm[62] for optimization. The optimization was terminated if there was no progress or when acceptable models had been found. We repeated the optimization process several times and generated in total 909722, 292151 and 201028 acceptable models for PTs with the most superficial, deepest and in-between BPs, respectively. Each of these models reproduced the characteristic electrophysiology of PTs, but utilized largely different ion channel distributions to do so (Supplementary Fig. S3a–f). A subset of models were additionally constrained to be able to perform coincidence detection multiple times within 300 ms. From each optimization run, we selected one model for which the maximal deviation from the mean of the empirical data across all objectives was minimal (0.9–1.9 mean STDs across objectives). In addition, from the whole database of models, we selected models that are most diverse in the active ion currents contributing to their $Ca^{2+}$ APs. To do so, we quantified the charge exchanged through active conductances for each model during the $Ca^{2+}$ AP and computed the contributions of different ion channels to the total hyperpolarizing or depolarizing currents. For each ion channel and each morphology, we selected the model in which the contribution of the respective channel was largest, which resulted in a set of models utilizing largely disparate mechanisms to generate dendritic $Ca^{2+}$ APs (Supplementary Fig. S3f). We hence selected 68 PT models for network embedding and simulations as described below (most superficial BP: 40, in-between BP: 13, deepest BP: 15).

## Network models

TC synapse positions along the PT models were measured as described above. Positions of excitatory synapses from cortical neurons were derived from an anatomically realistic network model of rat vS1[36], a procedure which we described in detail previously[32]. We registered the dendrites selected for multi-compartmental modeling in the network model at nine locations within the barrel column representing the C2 whisker, which is located approximately in the center of vS1, while preserving the PTs' in vivo soma depth[59]. The locations were the column center and equally spaced angular intervals with a distance of 100 μm to the column center. For each of the nine locations, we estimated the numbers and locations of cell type-specific synapses that impinge onto the dendrites of the PT models. Finally, inhibitory synapses were distributed randomly on the dendrites. Synapses were assigned to presynaptic neurons in the following manner. TC synapses: we randomly assigned each TC synapse to 350 TC neurons, the average number of neurons in the somatotopically aligned C2 barreloid[63]. Excitatory synapses: presynaptic neuron IDs are an output of the network model[36]. Inhibitory synapses: each synapse was assigned to one virtual neuron.

## Synapse models

Synapse models and parameters (rise/decay times, release probabilities, reversal potentials) were reported previously[32]. Briefly, conductance-based synapses were modeled with a double-exponential time course. Excitatory synapses contained both AMPA receptors (AMPARs) and NMDARs. Inhibitory synapses contained GABA$_A$Rs. The peak conductance of excitatory synapses from different presynaptic types was determined by assigning the same peak conductance to all synapses of the same type, activating all connections of the same type (i.e., all synapses originating from the same presynaptic neurons) one at a time, and comparing the simulated unitary postsynaptic potential (uPSP) amplitude distributions (mean, median and maximum) for a

fixed peak conductance with experimental measurements in vitro (cortical input[45]) or in vivo (TC input[29], Supplementary Fig. S3g). The peak conductance of each presynaptic type was then systematically varied until the squared difference between in silico and in vitro/vivo uPSP amplitude distributions were minimal (mean and median of the distributions were weighted twice relative to the maximum). We repeated this procedure for each multi-compartmental model using the connectivity model for the location in the center of the C2 column. The peak conductance at inhibitory synapses was fixed at 1 nS[64].

## Simulations

Simulations were performed using Python 3.8, dask[65] and NEURON 7.8[66]. First, we determined model configurations that result in simulations with AP rates that are within the range observed across PTs during ongoing periods preceding the whisker stimuli. For the 612 network-embedded PT models (i.e., 68 multi-compartmental models for 3 morphologies and 9 embeddings into the network model, respectively), we simulated prestimulus activity by activating neurons in the network model that are presynaptic to the PTs with the ongoing firing rates that we recorded in anesthetized animals across layers and for different excitatory (EXC) and inhibitory (INH) types, and in the VPM thalamus. We distributed 5000 inhibitory synapses uniformly across the dendrites and activated them with different multiples of the in vivo observed firing rates (i.e., 0.25 to 3.75 times the in vivo rates, in 0.25 steps). For each of these 9180 model configurations (i.e., 68 PT models * 9 network embeddings * 15 EXC/INH ratios), we simulated 48 trials of 1245 ms, of which we discarded the first 245 ms. 2241 of the 9180 configurations predicted ongoing AP rates as observed for PTs in vivo (i.e., > 0 Hz and ≤ 11 Hz). Second, we determined which of these 2241 model configurations result also in simulations with 1 AP, 2 AP and 3 AP burst rates that are within the respective ranges observed across PTs during the onset response evoked by whisker stimuli. For this purpose, we simulated 445 ms of ongoing activity as described above, and then activated neurons in the network model that are presynaptic to the PT models by generating Poisson spike trains for each presynaptic neuron based on the measured PSTHs of the respective types (Fig. 2e). We systematically tested whether our simulation results are robust with respect to different stimuli, different timings and ratios of sensory-evoked inhibition relative to excitation. Stimuli: principal whisker, three whiskers within the same row, or three whiskers within two adjacent rows, respectively. Timings: latencies as observed in vivo, inhibition shifted by − 1 or − 2 ms. Ratios: we activated inhibitory neurons with different multiples of the in vivo observed firing rates (i.e., 0.1 to 1.0 times the in vivo PSTHs, in 0.1 steps). For each of these 161352 model configurations (i.e., 2241 models with realistic ongoing rates * 3 stimuli * 3 EXC/INH timings * 8 EXC/INH ratios), we simulated 20 trials to select all configurations that predict sensory-evoked responses within the in vivo observed ranges (i.e., ≥ 1 AP probability: > 0%, 2 AP burst probability > 0 % and ≤ 40%, 3 AP burst probability ≥ 0% and ≤ 25%). 67424 of the 161352 configurations predicted pre- and post-stimulus activity as we had observed for PTs in vivo, out of which 22850 configurations contained sensory-evoked bursts. Third, we selected these 22850 model configurations and repeated the simulations, this time 200 instead of 20 trials per model configuration. Moreover, in the first coarse selection step, we had accepted 3 AP burst probabilities up to 25%. Now, we only selected those model configurations with 3 AP burst probabilities up to 13%, which is the maximal value observed. The simulations hence identified 20359 model configurations that predict pre- and post-stimulus activity, including sensory-evoked bursts of 2 and 3 APs, consistent with the in vivo data. These models comprised configurations for all three morphologies (most superficial BP: 12589, middle BP: 3298, deepest BP: 4472 models), all nine network locations, all 68 biophysical parameter sets, all 3 whisker stimuli, all INH timings, 7/8 post-stimulus EXC/INH ratios, and 13/15 pre-stimulus EXC/INH ratios. Finally, we used these 20359 model

configurations for in silico manipulations. For this purpose, we replayed the simulations multiple times, but inactivated sensory-evoked input from a presynaptic population. These manipulations were numbered as follows: (1) all EXC types in vS1 (which we refer to as indirect sensory input), (2) TC synapses from VPM thalamus (which we refer to as direct sensory input), (3) L2/3PNs, (4) L4SPs, (5) L5ITs, (6) L5PTs, (7) L6CCs, (8) VPM and L6CCs (Supplementary Fig. S5). For manipulation 9, we removed active conductances of the apical dendrite. We performed additional analyses models with the most superficial BP. To investigate distance-dependent impacts of TC synapse distributions, we selected models with 3 AP burst probability ≥ 2% (n = 880 model configurations), and simulated 1000 trials for each of them. We replayed the simulations 20 times, each time removing 50 TC synapse activations in 200 μm intervals with increasing soma distance ranging from 0-200 to 950-1150 μm (manipulation 10). We distinguished between sensory- and nonsensory-evoked $Ca^{2+}$ APs (Supplementary Fig. S10). For this purpose, we characterized the onset time (time of crossing − 30 mV), peak voltage and width of $Ca^{2+}$ APs (time between up and down crossing of − 30 mV) in trials which are abolished (no longer cross the − 30 mV) by removing indirect sensory input (i.e., sensory-evoked $Ca^{2+}$ APs) compared to those which remain (i.e., nonsensory-evoked $Ca^{2+}$ APs). Except for Supplementary Fig. S10, we did not analyze simulation trials with nonsensory-evoked $Ca^{2+}$ APs. We compared the waveforms of the sensory-evoked dendritic responses predicted in silico with those observed in vivo[46]. For this purpose, we computed the amplitude and duration (i.e., full-width half maximum) of the in silico responses that we identified as sensory-evoked $Ca^{2+}$ APs and those that were not (i.e., fast responses), and compared them with the data shown in Figs. 9 and 10 of Larkum & Zhu, 2002 (i.e., fast vs. complex dendritic potentials)[46]. We compared occurrences of $Ca^{2+}$ APs predicted in silico with those of in vivo observed dendritic $Ca^{2+}$ signals of PTs in the barrel cortex after whisker deflections. For this purpose, data reported by Takahashi et al., 2020[11] was reanalyzed as follows: sensory-evoked $Ca^{2+}$ signals from 229 PTs in 4 mice were detected during the 500 ms post-stimulus based on a threshold of three times the median absolute deviation. As for in silico analysis, trials were excluded if $Ca^{2+}$ signal onset was pre-stimulus (i.e., putative nonsensory-evoked $Ca^{2+}$ APs). To investigate the impacts of ongoing activity (manipulation 11), we selected all simulation trials with a burst response of 3 APs in the controls and then removed 50 synapse activations from EXC types in vS1 in the 20 ms window before TC onset, from different dendritic compartments (basal, trunk, apical tuft). To investigate the impact of increased ongoing activity, we selected all model configurations with 3 AP burst probability ≥ 1%, and simulated 200 trials for each where the in vivo recorded ongoing firing rates of EXC types were scaled by a factor from 0.5 to 2 throughout the simulation. As a proof of principle for how increased ongoing activity could increase the rate of triplets, we show results from models that had a triplet rate of at least 35% with a scaling factor of 1.7 applied to all types. Finally, plasticity has been reported for VPM synapses to PNs in deep layers of vS1, with ~1/3 of the synapses showing facilitation on a second stimulus in a series (class 1 C)[48]. To investigate whether synaptic facilitation in the TC→PT pathway could account for the observed bursts in our in vivo experiments (Supplementary Fig. S9), we repeated all simulations with an increased synaptic weight for all TC synapses (2.5 times the control value, as reported for the second stimulus[48]).

## Quantification and statistical analysis

All data are reported as mean ± standard deviation (STD) unless mentioned otherwise. All of the statistical details can be found in the figure legends, figures, and Results, including the statistical tests used, sample size n, and what the sample size represents (e.g., number of animals, number of cells). Significance was defined for p-values smaller than 0.05. Boxplots show median, 25% and 75%

percentile; for in vivo data, whiskers extend to 1.5 times the inter-quartile range, and data beyond whiskers are outliers; for in silico data, whiskers extend to the full data range, unless when shown side by side with in vivo data (i.e., whiskers then also extend to inter-quartile ranges, but due to the large in silico sample sizes, no outliers are shown to maintain visibility). We did all tests using the scipy Python package (version 0.18.2).

## Reporting summary

Further information on research design is available in the Nature Portfolio Reporting Summary linked to this article.

## Data availability

All data are publicly available. The neuroanatomical and electro-physiological data were deposited in Edmond: https://doi.org/10.17617/3.ST7EWL. A minimal simulation dataset and code to reproduce the simulations in this study is available at https://github.com/mpinb/in_silico_framework_hot_zone. The full simulation dataset (> 0.5 PB of data) can be made available upon request. Source data are provided in this paper.

## Code availability

All code and the custom-designed software to run simulations (In Silico Framework) are publicly available. The instructions for installing and running ISF, comprehensive documentation, tutorials and examples, all analyses routines, as well animated visualization of simulations and analyses are accessible via: https://wwwuser.gwdguser.de/-b.meulemeester/index.html. The code for generating, simulating and analyzing all multi-scale models is accessible for download via: https://github.com/mpinb/in_silico_framework_hot_zone. This repository comprises also a demo version of the code to reproduce the simulation and in silico manipulation examples in Figs. 4–6.

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

## Acknowledgements

We thank Bert Sakmann, Peter Strick and Idan Segev for comments, Martin Schwarz for providing the virus construct, Matthew Larkum and Naoya Takahashi for providing in vivo Ca$^{2+}$ data, Bjorge Meulemeester for documenting in silico routines, and Philipp Harth for visualizing cortex models. Funding was provided by the Max Planck Institute for Neurobiology of Behavior – Caesar (M.O.), European Research Council grants 633428, 101069192 (M.O.), Deutsche Forschungsgemeinschaft grants SFB 1089, SPP 2041 (MO), German Federal Ministry of Education and Research grant 01IS18052 (M.O.), Neuroscience Network North Rhine-Westphalia grant iBehave (M.O.), National Institute of Mental Health grant UM1MH130981-01 (C.P.J.d.K.), and NWO Open Competition grant ENW-M2, project OCENW.M20.285 (C.P.J.d.K.).

## Author contributions

M.O. conceived and designed the study. J.M.G. and A.B. contributed equally to this work and are listed alphabetically by last name. J.M.G. performed experiments and discovered the wiring specificity. A.B. performed simulations and discovered the TC coupling mechanism. J.M.G., A.B., R.F., and M.O. analyzed data. R.N. and C.P.J.d.K. contributed data. M.O. wrote the paper with A.B. and J.M.G.

## Funding

## Competing interests

The authors declare no competing interests.
