## [Transparent Peer Review file · Nature Communications]

Thalamus enables active dendritic coupling of inputs arriving at different cortical layers

Corresponding Author: Professor Marcel Oberlaender

Version 0:

Reviewer comments:

Reviewer #1

(Remarks to the Author)

The authors have done a very good job of addressing my previous comments, and the revised manuscript reads very well. A few outstanding issues:

* Previous Major 7 is still a point of confusion for me. Both the Larkum model and the conceptual model here involve "coincidence detection", and using "coincidence detection" to solely describe Larkum's version is perhaps the source of my confusion. I take the authors' point that the name "coincidence detection" might suggest a more precise timing of events than they believe exists here. However, I wonder if the critical difference is that, while Larkum argues backpropagating APs depolarize the initiation zone that generates the plateau potentials, the authors are arguing that in fact the depolarization is due to the direct thalamocortical synaptic input. (Both models seem like coincidence detection.) If my understanding is correct, perhaps they could simply edit the Discussion around coincidence detection to reflect their conclusion is about who is depolarizing the initiation zone (bAPs vs TC synapses).

* Related to this, lines 423-425 are difficult to believe as written: "First, direct sensory input from the thalamus initiates the sensory-evoked Ca²⁺ APs before the onset of the first sensory-evoked somatic APs, not after as in coincidence detection." The rates of Ca²⁺ APs, even during active behavior, when imaged in vivo seem too low for this to be true. If sensory-evoked thalamic activity were sufficient to trigger a Ca²⁺ AP, one would expect Ca²⁺ APs reliably on almost every single whisker stimulus. Minimally, the authors could compare the rate of Ca²⁺ APs in their simulations with published imaging experiments. I suspect these must be quite different, given what the authors write above.

Minor:

Figure 1. Why false color mCherry as green and Alexa-488 as red? I'm not sure the authors need to change this, but it was surprising.

Perhaps manipulations 9 and 10 should be mentioned in the legend of their relevant figure as the authors do elsewhere for other manipulations.

Fig.5b, 6a: Some of these traces are surprisingly short and do not look like plateau potentials. Are the authors sure a Ca AP occurred in all these cases?

(Remarks on code availability)

Reviewer #2

(Remarks to the Author)

The authors have made major efforts to address my comments and questions. They have included new data, focused the text and clarified the importance of investigating burst firing in the cortex. The finding of clustered innervation of an apical dendritic region by VPL axons is a noteworthy and novel result. This remains an important finding that I fully support for publication. Just a small point but there is a scale bar arrow in Fig 9e, I assume this should be a bar.

(Remarks on code availability)

Version 1:

Reviewer comments:

Reviewer #1

(Remarks to the Author)

The authors have addressed all my previous comments. I congratulate them on this study.

(Remarks on code availability)

I confirm that the authors have made their code available including documentation for installation and use. I have not attempted to install and execute the code, which I believe is not a reasonable request to make of referees.

Reviewer #1: The authors have done a very good job of addressing my previous comments, and the revised manuscript reads very well. A few outstanding issues:

*We thank the reviewer for the positive feedback on our revised manuscript. We addressed your remaining comments. In brief: Most importantly, we resolved your concern that we claim “sensory-evoked thalamic activity were sufficient to trigger a Ca^{2+} AP”. We apologize for our imprecise wording and corrected it. Moreover, we followed your advice to do a comprehensive comparison of our results with those obtained by others. We provide these **new data and analyses in new Fig. 5 and Fig. S6/7, and in three new result paragraphs**. These new data demonstrate full consistency of our *in silico* and *in vivo* results with those obtained via whole-cell recordings or Ca^{2+} imaging from the dendrites of PTs in the barrel cortex reported by the Larkum, Helmchen and Magee lab. For example, together with the Larkum lab, we reanalyzed data from Takahashi et al., Nat Neuro 2020 to obtain *in vivo* rates of whisker-evoked dendritic Ca^{2+} signals in PTs. Both their *in vivo* and our *in silico* data showed that **Ca^{2+} APs occur only in a small minority of trials** ($3\pm 5\%$ vs. $3\pm 4\%$), which is also in line with Xu et al., Nature 2012, who estimated an upper limit of 6%. Your advice helped us tremendously to further demonstrate the power of our approach, and to solidify our conclusions. For your convenience, we pasted in the revised text from the manuscript below (yellow).*

Major 1: Previous Major 7 is still a point of confusion for me. Both the Larkum model and the conceptual model here involve “coincidence detection”, and using “coincidence detection” to solely describe Larkum’s version is perhaps the source of my confusion. I take the authors’ point that the name “coincidence detection” might suggest a more precise timing of events than they believe exists here. However, I wonder if the critical difference is that, while Larkum argues backpropagating APs depolarize the initiation zone that generates the plateau potentials, the authors are arguing that in fact the depolarization is due to the direct thalamocortical synaptic input. (Both models seem like coincidence detection.) If my understanding is correct, perhaps they could simply edit the Discussion around coincidence detection to reflect their conclusion is about who is depolarizing the initiation zone (bAPs vs TC synapses).

We agree with the reviewer that our discussion of coincidence detection was still not sufficiently clear and even misleading. In fact, the reviewer summarized our results perfectly. We hence rewrote the discussion paragraph on bAP vs TC input inspired by this summary: “It is well known that PTs could utilize dendritic Ca^{2+} APs to perform coincidence detection – i.e., to multiplex inputs that impinge near simultaneously onto the basal and apical dendrites into bursts of somatic APs¹⁶. Currently, back-propagating APs (bAPs) are considered as the major mechanism that enables coincidence detection¹⁶. Our results suggest two major alterations of this perspective. First, we find that axons from primary thalamus target specifically and most densely the dendritic Ca^{2+} domain of PTs. We show that direct sensory input from the thalamus hence provides a weak depolarization to the Ca^{2+} domain that does not rely on bAPs. Second, in contrast to bAPs, we find that the weak depolarization of the Ca^{2+} domain by the thalamus precedes the first sensory-evoked somatic AP. We show that this fast activation of the Ca^{2+} domain widens the timing for coincidence detection to prestimulus periods – i.e., in addition to apical inputs that arrive after a sensory-evoked AP¹⁶, also those that arrive before it can contribute to the generation of sensory-evoked Ca^{2+} APs, and hence to the modulation of PT output with bursts. Taken together, our results reveal a novel mechanism by which PTs could utilize dendritic Ca^{2+} APs to perform coincidence detection, which we term TC coupling, and which enables pre- and poststimulus apical inputs to modulate the first sensory-evoked responses that leave the cortex with bursts. Our findings support the interpretation by Takahashi et al., that sensory-evoked Ca^{2+} signals in apical dendrites of PTs do not result from bAPs, but instead from a local activation of the Ca^{2+} domain¹² – i.e., via TC coupling.” Moreover, to increase clarity with respect to coincidence detection right from the start, we decided to modify our title to align it with the title

of Larkum et al., 1999. New title: *Thalamus enables active dendritic coupling of inputs arriving at different cortical layers*. Larkum's title: *A new cellular mechanism for coupling inputs arriving at different cortical layers*. We believe that our new title emphasizes that both studies propose mechanisms for coincidence detection – i.e., bAPs vs TC coupling. However, *in vivo* evidence in support of bAPs remains scarce. In fact, in Takahashi et al. 2020, Larkum concluded that not bAPs, but instead a local activation of the Ca^{2+} domain seems to trigger sensory-evoked Ca^{2+} signals. Here we reveal that direct TC input is an origin for this local activation, and we demonstrate that the available *in vivo* data supports TC coupling (see below).

Major 2: Related to this, lines 423-425 are difficult to believe as written: "First, direct sensory input from the thalamus initiates the sensory-evoked Ca^{2+} APs before the onset of the first sensory-evoked somatic APs, not after as in coincidence detection." The rates of Ca^{2+} APs, even during active behavior, when imaged *in vivo* seem too low for this to be true. If sensory-evoked thalamic activity were sufficient to trigger a Ca^{2+} AP, one would expect Ca^{2+} APs reliably on almost every single whisker stimulus. Minimally, the authors could compare the rate of Ca^{2+} APs in their simulations with published imaging experiments. I suspect these must be quite different, given what the authors write above.

*We agree that this would indeed be difficult to believe. We clarified this misleading sentence. Moreover, we had emphasized throughout the manuscript that TC input is too weak to drive Ca^{2+} APs, but instead, it allows additional inputs that impinge onto apical dendrites during and before a stimulus to be transformed into Ca^{2+} APs, and hence to evoke sensory-evoked bursts (e.g. as schematically shown in Fig. 8). However, we followed the reviewer's advice and now compared our *in silico* predictions with *in vivo* data from whole-cell recording and Ca^{2+} imaging of apical dendrites of PTs in barrel cortex by the Larkum, Helmchen and Magee labs. In fact, we contacted Larkum and reanalyzed with his team data from Takahashi et al., 2020. We provide these new data and analyses in new Fig. 5 and S6/7, and in 3 new result paragraphs: "In all PT models, sensory input from the whiskers triggered reliably dendritic potentials, which differed, however, in their waveforms (Fig. 5b). In the vast majority of the simulations (97% of 8,551,091 trials), sensory input failed to elicit dendritic Ca^{2+} APs. Instead, a fast rise and a quick decay phase characterized the waveform of these most frequently occurring sensory-evoked dendritic potentials (Fig. 5b, black trace). In the remaining trials, sensory input evoked dendritic Ca^{2+} APs that rose equally fast but decayed much slower with several peaks and dips (Fig. 5b, orange trace). Our *in silico* predictions are supported by *in vivo* dendritic recordings for the same experimental condition – i.e., for passive whisker deflections in anesthetized rats⁴⁶. In fact, the *in silico* predicted durations and amplitudes of both the fast potentials and the Ca^{2+} APs were virtually indistinguishable from those recorded *in vivo* near the primary BPs of PTs (Fig. 5c). This consistency is indeed remarkable, as we did not constrain, tune, or select the models to match these *in vivo* observations. Simultaneous recording and Ca^{2+} imaging of whisker-evoked responses near the primary BP provided further *in vivo* evidence in support of our *in silico* predictions¹⁸. These experiments demonstrated that sensory-evoked Ca^{2+} APs result generally in detectable Ca^{2+} signals, whereas fast potentials do not (Fig. S6). Consequently, we predict that whisker deflections evoke Ca^{2+} signals in only a small minority of trials ($3\pm 4\%$, $n=360$ PT models). Our analysis of *in vivo* Ca^{2+} signals in apical trunks of PTs ($3\pm 5\%$, $n=229$ PTs) – imaged upon whisker deflections by Takahashi et al.,¹¹ – supported this prediction (Fig. 5d). Taken together, consistent with *in vivo* recordings and Ca^{2+} imaging^{11, 18, 46, 47}, our simulations provided model consensus that sensory input evokes most frequently fast dendritic potentials, and in a small minority of trials Ca^{2+} APs."*

*"Sensory-evoked Ca^{2+} APs occurred in simulation trials with all somatic response types (Fig. 5e). These *in silico* predictions are supported by *in vivo* experiments that combined somatic recordings with dendritic Ca^{2+} imaging – i.e., the study observed Ca^{2+} signals during responses with 1 AP, or with bursts of 2 or 3 APs¹⁸. Both *in vivo* and *in silico* results hence indicate that observing a Ca^{2+} AP is generally insufficient*

to infer the response that sensory input will evoke at the soma (**Fig. 5f**). However, the *in vivo* data seems to suggest that the inverse may be possible – i.e., to infer the occurrence of a Ca^{2+} AP from the somatic response¹⁸. Ca^{2+} signals occurred reliably during bursts of 3 APs, whereas responses with 1 AP or with bursts of 2 APs occurred also in the absence of Ca^{2+} signals¹⁸. Our simulations supported this interpretation. Across PT models, virtually all trials with bursts of 3 APs had Ca^{2+} APs (**Fig. 5g**). In contrast, a vast majority of trials with somatic responses of 1 AP, or bursts of 2 APs had no Ca^{2+} APs, but instead fast potentials as their dendritic response (**Fig. S7**). Thus, supported by simultaneous *in vivo* recordings and Ca^{2+} imaging¹⁸, our simulations provided model consensus that a burst of 3 APs is generally sufficient to infer the occurrence of a sensory-evoked Ca^{2+} AP (**Fig. 5f**).”

“We chose this experimental condition, because such ‘active touches’ were reported to lead to a ~3-fold increase of the sensory-evoked Ca^{2+} signals in apical dendrites of PTs, likely due to additional input to upper layers of the barrel cortex from whisker-related activity in motor cortex⁴⁷. According to TC coupling, a 3-fold increase of sensory-evoked Ca^{2+} signals (i.e., putative Ca^{2+} APs¹⁸) should facilitate specifically the occurrences of bursts of 3 APs. This was indeed the case. We found that active touches evoked the same three response types (**Fig. 10b**) that we had observed for whisker deflections (i.e., passive touches). In fact, PTs responded to active touches with 1 AP ($p=0.48$) or bursts of 2 APs ($p=0.45$) as frequently as they did to passive touches, whereas bursts of 3 APs occurred on average 3.7 times more frequently upon active touches ($p=0.02$). Thus, active and passive touches evoked virtually identical responses across PTs (Mann-Whitney U: $p=0.97$), with the exception of bursts of 3 APs (**Fig. 10c**). Moreover, compared to periods of quiescence and whisking (**Fig. 10d**), occurrences of bursts of 3 APs increased upon active touch (ANOVA with multiple comparison: $p=0.02$), whereas occurrences of 1 AP ($p=0.65$) or bursts of 2 APs ($p=0.08$) did not. This result supports observations by Takahashi et al., which showed that ‘active behavioral engagement’ also increases occurrences of sensory-evoked Ca^{2+} signal in PTs¹¹. Taken together, experimental conditions with increased occurrences of sensory-evoked Ca^{2+} signals in apical dendrites^{11, 12, 47} also increase specifically and to similar degrees occurrences of sensory-evoked bursts of 3 APs. These results provide further *in vivo* evidence in support of TC coupling.”

Minor 1: Figure 1. Why false color mCherry as green and Alexa-488 as red? I’m not sure the authors need to change this, but it was surprising.

In our view, coloring the smaller structures (TC boutons) in green and larger structures (dendrites) in red increased visibility. We added a statement about the color maps to the figure legend.

Minor 2: Perhaps manipulations 9 and 10 should be mentioned in the legend of their relevant figure as the authors do elsewhere for other manipulations.

We added this information to the legend of Fig. 4.

Minor 3: Fig.5b, 6a: Some of these traces are surprisingly short and do not look like plateau potentials. Are the authors sure a Ca AP occurred in all these cases?

*The reviewer is correct that the durations of sensory-evoked dendritic depolarizations are highly variable. In new **Fig. 5 and S7**, we now show histograms for the *in silico* durations of dendritic depolarizations for responses of 0, 1, 2 and 3 APs, respectively. The histograms show clearly that the depolarizations occur in two distinct forms: fast potentials and, when they reach threshold, Ca^{2+} APs. These *in silico* predictions are fully in line with the whole-cell recordings that Larkum & Zhu performed near the primary BP of PTs under the same experimental condition investigated here. Larkum & Zhu also observed Ca^{2+} APs, which they called ‘complex’ potentials, with high variability in duration, including shorter ones (<20 ms) that resemble those spotted by the reviewer in our *in silico* examples. In fact, for the vast majority (~99%) of*

trials with bursts of 3 APs, our simulations predict Ca^{2+} APs with waveforms that match the classification of complex potentials by Larkum & Zhu (**Fig. S7**). For virtually none of the trials with bursts of 3 APs, our simulations predict waveforms that match the classification of fast potentials by Larkum & Zhu (note: fast potentials dominate the trials with 0, 1 and 2 APs, **Fig. S7**). However, the *in vivo* data by Larkum & Zhu suggested that there might be a third form of dendritic potentials, which they termed 'slow'. These slow potentials occurred very rarely *in vivo* and had durations longer than those of fast and shorter than those of complex potentials. Our simulations also support this *in vivo* observation. During bursts of 3 APs, our simulations predicted very rarely (~1% of the trials) waveforms that matched the classification of slow potentials by Larkum & Zhu (**Fig. S7**). Here, we did not consider these slow potentials as Ca^{2+} APs. This detailed *in vivo* vs. *in silico* comparison of dendritic waveforms is provided in the legend of **Fig. S7**.

Reviewer #2: The authors have made major efforts to address my comments and questions. They have included new data, focused the text and clarified the importance of investigating burst firing in the cortex. The finding of clustered innervation of an apical dendritic region by VPL axons is a noteworthy and novel result. This remains an important finding that I fully support for publication.

We thank the reviewer for the positive feedback on our revised manuscript, the constructive comments in the previous revision round, and for supporting the publication of our study.

Minor 1: Just a small point but there is a scale bar arrow in Fig 9e, I assume this should be a bar.
We have removed the arrow.